# Automatic Construction of Clinical Scoring Systems with LLM Agents

**Silas Ruhrberg Estévez** [1]    **Christopher Chiu** [1]    **Mihaela van der Schaar** [1]

## Abstract

Modern clinical practice relies on evidence-based guidelines implemented as compact scoring systems composed of a small number of interpretable decision rules. While machine-learning models achieve strong performance, many fail to translate into routine clinical use due to misalignment with workflow constraints such as memorability, auditability, and bedside execution. We argue that this gap arises not from insufficient predictive power, but from optimizing over model classes that are incompatible with guideline deployment. Deployable guidelines often take the form of unit-weighted clinical checklists, formed by thresholding the sum of binary rules, but learning such scores requires searching an exponentially large discrete space of possible rule sets. We introduce `AgentScore`, which performs semantically guided optimization in this space by using LLMs to propose candidate rules and a deterministic, data-grounded verification-and-selection loop to enforce statistical validity and deployability constraints. Across eight clinical prediction tasks, `AgentScore` outperforms existing score-generation methods and achieves AUROC comparable to more flexible interpretable models despite operating under stronger structural constraints. On two additional externally validated tasks, `AgentScore` achieves higher discrimination than established guideline-based scores.

## 1. Introduction

Clinical decision-making is inherently difficult, requiring clinicians to act under uncertainty, time pressure, and incomplete information (Patton, 1978). Over recent decades, clinical guidelines have pushed medicine toward evidence-based care, moving beyond the opinions of individual clinicians

(Sur & Dahm, 2011; Wieten, 2018). A central instrument in this shift is the clinical scoring system: a compact set of explicit rules mapping a small number of routinely available patient measurements to risk strata or management recommendations (Challener et al., 2019). When well designed, such scores standardize decisions, support resource allocation, and facilitate communication across care settings (Woolf et al., 1999). From a machine learning perspective, these artifacts are best viewed not as approximate regressors, but as a deliberately constrained model family optimized for bedside execution, recall, and auditability (Ustun & Rudin, 2015; Zhang et al., 2021).

Despite their ubiquity, effective scoring systems such as CURB-65 (Lim et al., 2003) must satisfy stringent practical requirements (Desai & Gross, 2019; Moons et al., 2015). They must rely on routinely available inputs, generalize across institutions despite missingness and measurement shift (Dambha-Miller et al., 2020), and remain interpretable and memorable for reliable bedside recall and audit without computational aids (Graham et al., 2011). Meeting these requirements in practice remains challenging. Most widely used scores are derived from expert consensus or manual analysis of observational studies (Woolf et al., 1999), often via regression models discretized for bedside use (see Fig. 1) (Sullivan et al., 2004). This process is labor-intensive, slow to update, and can yield brittle feature and threshold choices that are difficult to revise (Wasylewicz & Scheepers-Hoeks, 2018; Woolf et al., 1999).

In parallel, increasingly complex machine learning models have achieved strong performance on clinical prediction tasks (Takita et al., 2025; Killock, 2020; Shickel et al., 2018). However, even when accurate and ostensibly interpretable, many such models remain poorly matched to guideline deployment: they rely on inputs that are inconsistently available, require preprocessing and software-mediated inference, and produce continuous-weight computations or decision thresholds that are difficult to execute and audit reliably at the bedside (Topol, 2019; Chen et al., 2021). This misalignment is reinforced by optimization incentives. Unconstrained models admit smooth parameterizations and efficient gradient-based training, whereas enforcing checklist structure induces a discrete subset selection problem with hard constraints on size and rule form, yielding a nonconvex and often NP-hard objective (Ustun & Rudin, 2015;

---

[1]DAMTP, University of Cambridge, Cambridge, UK. Correspondence to: Silas Ruhrberg Estévez <sr933@cam.ac.uk>.

*Proceedings of the 43rd International Conference on Machine Learning*, Seoul, South Korea. PMLR 306, 2026. Copyright 2026 by the author(s).

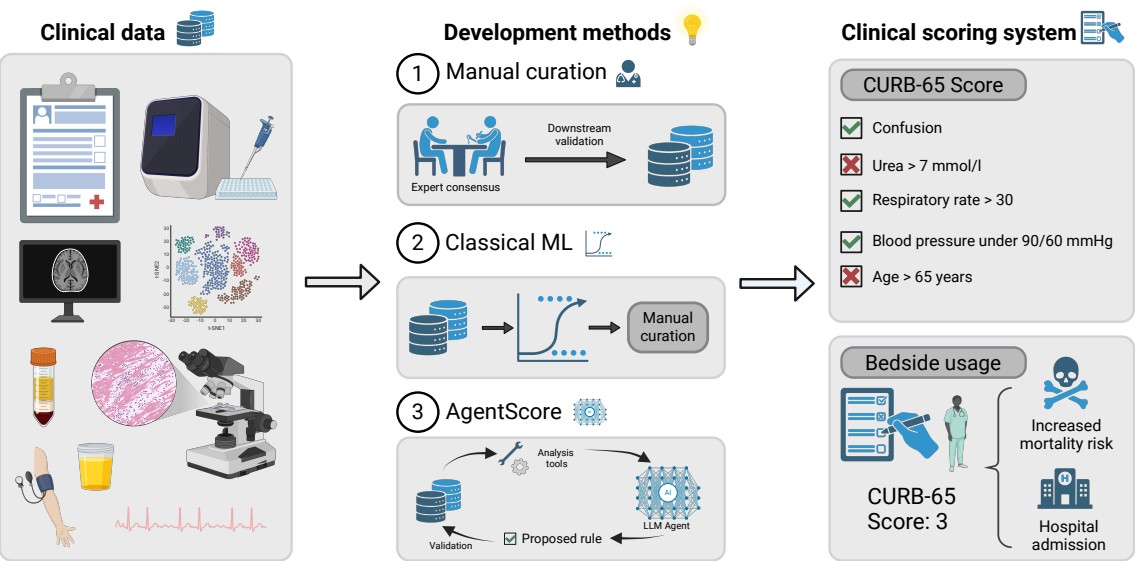

*Figure 1.* **Clinical scoring systems:** Clinical scoring systems translate routinely collected patient information into compact, explicit checklists that can be applied, recalled, and audited at the bedside. Traditional development relies on expert consensus, manual curation, or post-hoc simplification of classical machine-learning models. In contrast, `AgentScore` uses LLM-guided rule proposal together with deterministic, data-grounded validation to construct unit-weighted checklist scores.

Bertsimas & Stellato, 2020). As a result, much of modern clinical ML implicitly optimizes over model classes that are convenient to train but incompatible with guideline work-flows (Shortliffe & Sepulveda, 2018), while simple scores integrate directly into clinical practice (Graham et al., 2011).

Recent machine-learning work has begun to automate the learning of sparse clinical scores and unit-weighted checklist models, including mixed-integer optimization approaches that select compact checklists over fixed candidate features (Zhang et al., 2021). These methods show that checklist-style models can be learned automatically and can perform competitively with more flexible interpretable models. However, they typically assume that the candidate rule space has already been constructed.

**Gap.** Current clinical ML pipelines often treat deployability as a downstream engineering consideration rather than a first-class modeling constraint (Kelly et al., 2019). While recent work emphasizes *interpretability*, **interpretability does not guarantee deployability** (Lipton, 2018): a decision tree or sparse logistic regression may be human-understandable, but in its native form it typically cannot be executed and audited reliably at the bedside (e.g., floating-point arithmetic, non-memorable thresholds). Conversely, existing integer risk scores and checklist learning methods largely optimize over a fixed, pre-constructed feature matrix (Liu et al., 2022; Zhang et al., 2021) and therefore do not address the setting where clinically meaningful derived rules must be constructed from raw variables. A gap therefore remains for methods that jointly optimize *rule construction* and *checklist-compatibility constraints* by design.

**Contributions**

**Conceptual.** We formalize deployable clinical checklist learning as a joint combinatorial optimization problem over rule construction and subset selection, rather than checklist selection over fixed pre-constructed features, within a constrained model class of unit-weighted checklists. The rule language supports temporal patterns, physiologic ratios, and shallow compositions, explicitly encoding bedside cognitive and operational constraints.

**Algorithmic.** We introduce `AgentScore`, a framework that bridges LLMs and discrete optimization: LLMs propose semantically structured candidate rules within a restricted clinical grammar, while deterministic tools enforce grammar validity, statistical screening, redundancy control, and checklist-size constraints.

**Empirical.** Across eight clinical prediction tasks spanning MIMIC-IV and eICU, `AgentScore` matches or exceeds state-of-the-art integer score and checklist learning baselines despite operating under stricter structural constraints, and on two externally validated tasks it achieves higher discrimination than established guideline-based scores while remaining suitable for manual execution.

## 2. Deployable Clinical Scoring Systems

We treat bedside deployability as a hard modeling primitive rather than an auxiliary constraint. Once specified, the score can be applied as a short bedside checklist without mandatory software-mediated inference, floating-point computation, or access to the training pipeline. Accordingly, we restrict the hypothesis class itself to scoring systems

that are natively compatible with clinical guidelines and routine bedside execution. Under this view, a clinical score is not an approximation to a continuous predictor, but a discrete decision object: a sparse, unit-weighted collection of human-interpretable binary rules (Zhang et al., 2021). Its structure is dictated by operational and cognitive constraints, including memorability, auditability, and arithmetic-free use, rather than by predictive expressiveness alone.

**Problem setting.** Let $\mathcal{D} = \{(\mathbf{X}_i, y_i)\}_{i=1}^N$ denote a dataset of patient trajectories, where $\mathbf{X}_i \in \mathbb{R}^{p \times T_i}$ is a matrix of $p$ clinical variables observed over $T_i$ timepoints for patient $i$, and $y_i \in \{0, 1\}$ is a binary outcome associated with each trajectory. For deployment, we compute a fixed, deployable representation $\tilde{\mathbf{x}}_i = \phi(\mathbf{X}_i) \in \mathbb{R}^{p'}$ that includes both static summaries and predefined temporal transformations; rules operate on $\tilde{\mathbf{x}}_i$.

**Rule-based feature construction.** A *rule* is a binary-valued predicate $r(\tilde{\mathbf{x}}) : \mathbb{R}^{p'} \to \{0, 1\}$, corresponding to a clinical statement, drawn from a restricted, clinically motivated rule language. The candidate rule dictionary is $\mathcal{R} = \{r_j(\tilde{\mathbf{x}})\}_{j=1}^{|\mathcal{R}|}$. The clinical checklists and the allowed rule families reflect constructs recurring in guideline-based scoring systems and bedside decision rules (see Appendix A).

**Why unit-weighted checklists?** Clinical scoring systems are typically applied under time pressure, interruption, and incomplete information, where cognitive load and memorability are primary bottlenecks (Miller, 1956). Empirically, widely adopted clinical guidelines almost universally take the form of short additive checklists composed of binary conditions and small total score ranges. Classical results on *improper linear models* further show that equal-weighted additive models often perform comparably to optimally weighted linear predictors (Dawes, 1979), aligning with evidence that simple, transparent heuristics are particularly effective under time pressure and uncertainty (Gigerenzer & Gaissmaier, 2011). Furthermore, prior work has shown that automatically constructed checklist models can achieve performance comparable to more flexible models when operating over fixed feature representations (Zhang et al., 2021). Allowing non-unit weights increases mental arithmetic burden and reduces transparency, often necessitating calculators or electronic support. Similarly, deeply nested logic trees require tracking multiple contingent branches and intermediate states, which exceeds cognitive limits in bedside settings; as a result, such models are rarely adopted without computational mediation. We therefore adopt simplicity as an explicit design objective rather than a byproduct of regularization. By deliberately restricting the model class to the simplest structure, unit-weighted $N$-of-$M$ checklists, we maximize memorability, auditability, and reliable bedside execution.

**Rule Language for Deployable Scoring Systems**
**(i) Numeric threshold rules.**
$$r(\tilde{\mathbf{x}}) = \mathbb{I}[\tilde{x}_k \odot c], \qquad \odot \in \{>, \geq, <, \leq\}, \ c \in \mathbb{R}.$$

**(ii) Numeric range rules.**
$$r(\tilde{\mathbf{x}}) = \mathbb{I}[c_{\text{low}} \leq \tilde{x}_k \leq c_{\text{high}}].$$
**(iii) Categorical inclusion rules.**
$$r(\tilde{\mathbf{x}}) = \mathbb{I}[\tilde{x}_k \in \mathcal{A}].$$
**(iv) Binary presence rules.**
$$r(\tilde{\mathbf{x}}) = \mathbb{I}[\tilde{x}_k = 1].$$
**(v) Physiological ratio and contrast rules.**
$$r(\tilde{\mathbf{x}}) = \mathbb{I}[g(\tilde{\mathbf{x}}) \odot c], \qquad g(\tilde{\mathbf{x}}) \in \left\{ \frac{\tilde{x}_a}{\tilde{x}_b}, \ \tilde{x}_a - \tilde{x}_b \right\},$$
**(vi) Count-based rules.**
$$r(\tilde{\mathbf{x}}) = \mathbb{I}\left[ \sum_{j \in \mathcal{J}} r_j(\tilde{\mathbf{x}}) \geq m' \right].$$
**(vii) Logical composition rules.** Given base rules $r_\ell{}_{\ell=1}^L$, we allow shallow Boolean compositions (AND/OR). We define $\text{Depth}(r)$ as the number of logical operations and restrict it to preserve bedside memorability:
$$r(\tilde{\mathbf{x}}) = \begin{cases} \bigwedge_{\ell=1}^L r_\ell(\tilde{\mathbf{x}}) & \text{(AND)}, \\ \bigvee_{\ell=1}^L r_\ell(\tilde{\mathbf{x}}) & \text{(OR)}. \end{cases}$$
**(viii) Temporal and distributional rules.** Temporal rules operate on predefined summaries of longitudinal measurements included in $\tilde{\mathbf{x}}_i = \phi(\mathbf{X}_i)$: $r(\tilde{\mathbf{x}}_i) = \mathbb{I}[h(\mathbf{X}_i) \odot c]$.
$$h(\mathbf{X}_i) \in \left\{ \Delta x_k^{(t)}, \frac{x_k^{(t)} - x_k^{(0)}}{x_k^{(0)}}, \max_t x_k^{(t)} - \min_t x_k^{(t)} \right\},$$

In addition, we allow distributional normalizations, including quantile-based rules $\mathbb{I}[\tilde{x}_k \odot Q_k(q)]$, standardized rules $\mathbb{I}[(\tilde{x}_k - \mu_k)/\sigma_k \odot z]$, and percent-change rules $\mathbb{I}[\Delta x_k / x_k \odot \gamma]$.

**Score definition.** A clinical scoring system selects a subset $\mathcal{S} \subseteq \mathcal{R}$ of at most $m$ rules and assigns each a unit weight: $S(\tilde{\mathbf{x}}) = \sum_{r_j \in \mathcal{S}} r_j(\tilde{\mathbf{x}})$, yielding a discrete score $S(\tilde{\mathbf{x}}) \in \{0, 1, \ldots, m\}$. An integer threshold $\tau \in \{0, \ldots, m\}$ induces a binary clinical decision rule:
$$\widehat{y}(\tilde{\mathbf{x}}) = \begin{cases} 1, & S(\tilde{\mathbf{x}}) \geq \tau, \\ 0, & \text{otherwise}. \end{cases}$$

The resulting model is a unit-weighted $N$-of-$M$ checklist, with $M = |\mathcal{S}|$ denoting the number of rules and $N = \tau$ the decision threshold.

**Optimization objective.** We seek guideline-style scoring systems that maximize empirical clinical utility under deployability constraints:

$$\max_{\mathcal{S} \subseteq \mathcal{R}} \quad \mathcal{U}(S(\tilde{\mathbf{x}}), y; \mathcal{D}_{\text{val}})$$

$$\text{s.t.} \quad |\mathcal{S}| \leq m, \; \text{Depth}(r) \leq d \quad \forall r \in \mathcal{S}. \tag{1}$$

Here $\mathcal{U}$ denotes an empirical, non-convex utility (e.g., AU-ROC, net benefit, or decision-curve utility) evaluated on held-out data. This optimization is NP-hard and non-differentiable due to discrete rule selection and combinatorial structural constraints (Ustun & Rudin, 2015). Unlike continuous relaxations or surrogate losses, we directly optimize the target utility via constrained, agent-guided search, ensuring that deployability constraints are enforced throughout.

**Definition 2.1** (Deployable Clinical Scoring System). A *deployable clinical scoring system* is a tuple $\mathcal{G} = (\mathcal{S}, S, \tau)$, where $\mathcal{S} \subseteq \mathcal{R}$ is a finite set of binary, human-interpretable rules, $S(\tilde{\mathbf{x}}) = \sum_{r_j \in \mathcal{S}} r_j(\tilde{\mathbf{x}})$ is an integer-valued score, and $\tau$ is a decision threshold. The system is *deployable* if it satisfies:

1. **Parsimony:** $|\mathcal{S}| \leq m$ for small $m$.

2. **Interpretability:** Each rule corresponds to a clinically meaningful statement over routinely collected variables.

3. **Memorability:** Rules are binary, unit-weighted, and shallowly composed, enabling reliable recall and manual application.

4. **Operational deployability:** The score can be evaluated manually at the point of care from routinely available variables, without model-serving infrastructure.

5. **Predictive adequacy:** The model achieves acceptable discrimination and calibration on the target population, subject to the above constraints.

**Example: UK CF Registry 1-year Mortality Score.** `AgentScore` produces compact, guideline-style checklists. Table 1 shows a representative one-year mortality score whose rules span multiple allowed rule families and are constructed by `AgentScore` rather than taken directly from raw dataset columns.

*Table 1.* **UK CF Registry three-year mortality checklist generated by `AgentScore`.** Each satisfied rule contributes one point.

| Checklist rule (satisfied?) | Points |
|---|---|
| $\text{FEV}_1$ predicted $\geq 15\%$ decline from 5-year best | $+1$ |
| Current $\text{FEV}_1$ predicted $\leq 50\%$ | $+1$ |
| Current $\text{FEV}_1$ predicted $\leq 30\%$ | $+1$ |
| IV antibiotics: $\geq 15$ days (hospital) **or** $\geq 20$ days (home) | $+1$ |
| **High-risk threshold** | $\mathbf{S(\tilde{x}) \geq 2}$ |

## 3. Related Work

Our work lies at the intersection of (i) interpretable machine learning, (ii) sparse clinical score learning, and (iii) LLM-guided rule generation. Extended comparisons are provided in Appendix C.

**General interpretable models in healthcare.** A broad class of methods targets predictive performance while maintaining transparency. Generalized additive models (GAMs) such as Explainable Boosting Machines (EBMs) (Hastie & Tibshirani, 1986; Nori et al., 2019) and sparse linear models (e.g., LASSO) provide intelligibility via shape functions or feature selection, and are often used as pragmatic baselines on tabular clinical data. However, *interpretability* alone does not ensure reliable, unaided bedside deployment (Lipton, 2018). These models typically yield continuous coefficients (e.g., $0.41 \times \text{Age}$), require non-trivial arithmetic, or rely on look-up tables for non-linear terms, making unaided manual use difficult at the point of care. Moreover, they do not typically enforce the operational constraints of guideline artifacts, such as bounded score ranges, unit weights, and unordered checklist execution. Rule-based models such as CORELS (Angelino et al., 2018), optimal decision trees including GOSDT (Hu et al., 2019), and Bayesian Rule Lists (Letham et al., 2015) offer alternative forms of interpretability. However, these models rely on ordered or hierarchical evaluation, in contrast to the unordered, additive checklist structure typical of clinical scoring systems. We also include heuristic scoring pipelines such as AutoScore (Xie et al., 2020). While AutoScore produces integer-valued point systems, it often produces multi-bin point systems with wider score ranges and non-unit weights, which can increase cognitive load compared with short unit-weighted checklists (Graham et al., 2011).

**Sparse clinical score learning.** Distinct from general interpretable ML, score-learning methods aim to produce compact point-based instruments that can be executed manually at the bedside. Integer-weighted approaches such as RiskSLIM (Ustun & Rudin, 2015) and FasterRisk (Liu et al., 2022) formulate score construction as sparse integer optimization, yielding concise linear models with small integer coefficients and strong face validity. These methods improve on classical regression-to-points pipelines, which discretize and round regression coefficients into integer scores, but they still often require non-unit or signed weights. Even short scores can therefore impose operational burden: adding terms such as $+5$ and $-3$ increases mental arithmetic demands and may raise the risk of error under time pressure. A related line of work directly learns unit-weighted checklists. For example, Optimal Checklists (Zhang et al., 2021) formulates checklist learning as mixed-integer optimization over a fixed set of candidate features, producing $N$-of-$M$

*Table 2.* **Comparison of representative interpretable score-learning approaches.** Prior methods optimize linear, additive, or sequential rule models over fixed feature encodings. In contrast, `AgentScore` targets deployable-by-design, unit-weighted checklist scores by searching a guideline-compatible rule language under hard structural constraints.

| Method | Score form | Weights | Rule language | Search / construction | Derived rule construction | Unit-weighted unordered checklist |
|---|---|---|---|---|---|---|
| RiskSLIM | linear score | non-unit integers | binned thresholds / fixed binaries | MIP over coefficients | × | × |
| FasterRisk | linear score | non-unit integers | binned thresholds / fixed binaries | approximate integer optimization | × | × |
| AutoScore | additive score | non-negative integers | binned thresholds | pipeline (rank $\rightarrow$ bin $\rightarrow$ score) | × | × |
| CORELS | rule list | N/A (sequential logic) | binarized features / thresholds | branch-and-bound search | × | × |
| Optimal Checklists | $N$-of-$M$ checklist | unit (0/1) | binned thresholds / fixed binaries | MIP over rule selection | × | ✓ |
| `AgentScore` | $N$-of-$M$ checklist | **unit** (0/1) | **clinical rules / thresholds** | **constrained search + validation** | ✓ | ✓ |

rules without coefficient arithmetic. However, such methods primarily select among pre-computed columns; they do not address the joint problem of constructing and selecting derived clinical rules. Semantic rules such as physiologic ratios, temporal changes, percentile-based thresholds, and logical compositions must therefore be manually engineered *a priori*. This leaves rule design as a prerequisite rather than part of the learning problem, limiting coverage of the combinatorial rule space that clinical guideline authors routinely consider. In contrast to coefficient-learning approaches, our target object is an unordered, unit-weighted clinical checklist in which the central challenge is not optimizing numerical weights, but discovering, validating, and assembling clinically meaningful rules.

**LLM-guided rule and feature generation.** Large language models have been explored for structured discovery, including feature generation, program synthesis, and guidance in combinatorial search spaces (Nam et al., 2024; Balek et al., 2025; Liu et al., 2025). In clinical settings, prior work reports limitations in robustness, calibration, and guideline adherence when LLMs are used as standalone decision-makers (Williams et al., 2024; Artsi et al., 2025). We instead constrain LLMs to a narrow, verifiable role: proposing candidate rules within a restricted, guideline-compatible language that encodes admissible rule families and shallow compositions. All proposals are subsequently screened and selected using held-out patient data under explicit deployability constraints (unit weights, bounded size, limited depth), so LLMs act as semantic proposal mechanisms embedded in a deterministic, data-grounded optimization loop rather than as autonomous decision-makers. In this design, the LLM does not determine the final score, assign risk, or validate clinical utility; it only expands the space of candidate rules that can then be accepted or rejected by transparent statistical criteria. This separation of semantic proposal from statistical verification yields reproducible rule selection and ensures that every retained rule is both interpretable and empirically supported. As longitudinal EHR signals become increasingly available, the bottleneck shifts from data scarcity to synthesizing deployable rules from routine measurements; we address this by combining LLM-guided proposal over a guideline-compatible rule language with deterministic, data-grounded validation under hard checklist constraints.

## 4. `AgentScore`

**Problem setup.** Learning deployable clinical scoring systems imposes three competing requirements. First, candidate rules must express clinically meaningful semantics, including derived quantities, temporal patterns, and shallow compositional constructs that are rarely available as explicit features in clinical datasets. Second, rule generation must be strictly data-grounded and robust to hallucination, bias, and spurious correlations, with every decision evaluated under explicit, verifiable metrics. Third, the resulting scores must satisfy hard deployability constraints while retaining acceptable predictive utility. Formally, we seek to learn a unit-weighted checklist rule set $\mathcal{S} = \{r_1, \ldots, r_{|\mathcal{S}|}\} \subseteq \mathcal{R}$ from a structured rule space $\mathcal{R}$ that maximizes predictive utility $\mathcal{U}(\mathcal{S})$ subject to hard constraints on rule structure and checklist size, i.e., $|\mathcal{S}| \leq m$. This induces a constrained subset selection problem over $\mathcal{R}$ whose size grows combinatorially with grammar depth, making exhaustive enumeration and direct optimization over the full derived-rule space impractical at the required level of expressiveness (see Appendix E).

**LLM usage: proposal vs. verification.** Injecting semantic understanding and domain knowledge into rule construction naturally motivates the use of LLMs. However, direct and unconstrained application of LLMs is unsuitable in medical settings: free-form generation provides no guarantees of validity or empirical grounding and introduces significant hallucination risks. Conversely, existing score-learning pipelines operating over fixed feature encodings are limited to pre-specified thresholds and binned variables, and cannot systematically discover higher-level clinical rules without exponential enumeration or extensive manual feature engineering. We therefore introduce `AgentScore`, a framework that treats LLMs not as a decision-makers, but as a *structured semantic proposal mechanism* embedded within a deterministic, tool-mediated evaluation loop. This design enables tractable search over an expressive, guideline-compatible hypothesis class while enforcing data grounding via tool verification and deployability by construction. Algorithmically, `AgentScore` instantiates an LLM-assisted combinatorial optimization procedure in which semantic exploration is decoupled from acceptance and optimization, yielding auditable, data-driven rule induction.

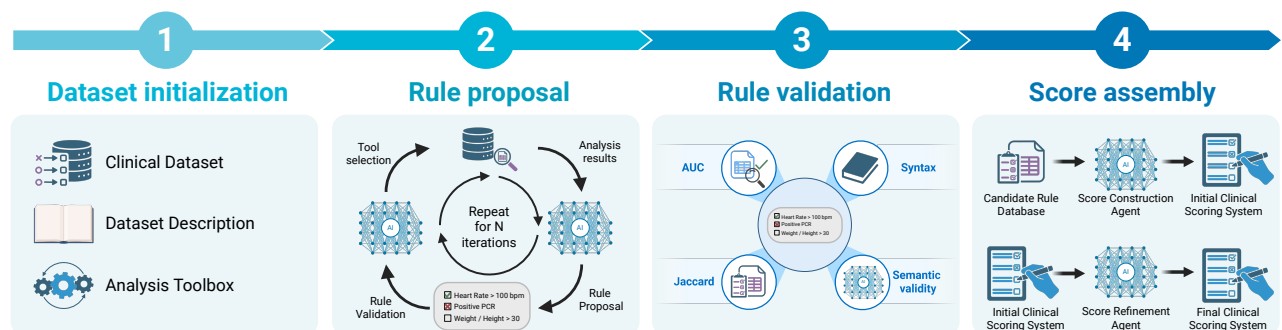

*Figure 2.* **Overview of `AgentScore`.** An LLM-based proposal agent generates candidate rules from a dataset description and tool-mediated aggregate statistics; it never receives patient-level records. Proposed rules are screened by a deterministic validation module enforcing statistical performance and grammar-level deployability constraints (e.g., complexity limits, unit weights). Statistically admissible rules are reviewed by a clinical plausibility agent to filter semantically incoherent proposals, after which a score assembly agent selects a compact clinical checklist from the retained pool and the evaluation tools choose the operating threshold.

**Overview.** `AgentScore` approximates the resulting checklist learning problem via a decomposition into three stages: *rule proposal*, *rule filtering*, and *score assembly* (Figure 2; Algorithm 1). Each stage enforces a distinct subset of constraints, enabling tractable optimization over the guideline-compatible rule space. Algorithm 1 provides the main learning procedure, while full pseudocode, tool interfaces, and prompts are provided in Appendix F and Appendix G.

---

**Algorithm 1** `AgentScore` learning procedure

**Require:** Data $(X, y)$, task description $\mathcal{T}$, rule budget $m$, thresholds $\tau_{\text{rule}}, \delta$
**Ensure:** Checklist $\mathcal{S}$ and threshold $\tau$
1: Initialize rule pool $\mathcal{P} \leftarrow \emptyset$ and tool interface $\mathcal{I}$
2: **for** each proposal round **do**
3:     LLM proposes candidate rules $C$ from grammar $\mathcal{R}$
4:     **for** each $r \in C$ **do**
5:         **if** $\text{AUROC}(r) \geq \tau_{\text{rule}}$ and $\max_{r' \in \mathcal{P}} J^+(r, r') \leq \delta$ **then**
6:             Add $r$ to $\mathcal{P}$ if clinically plausible
7:         **end if**
8:     **end for**
9: **end for**
10: Assemble checklist $\mathcal{S} \subseteq \mathcal{P}$ with $|\mathcal{S}| \leq m$
11: Iteratively refine $\mathcal{S}$ using feedback from data
12: Select decision threshold $\tau$ from data

---

**Rule proposal.** Candidate rules are proposed by a language-model agent constrained to emit structured rules from a predefined grammar (Appendix A). The agent does not observe patient-level records and interacts with the dataset exclusively through a fixed tool interface exposing feature metadata and aggregate evaluation statistics. This design limits direct exposure of sensitive data, but it is not a formal privacy guarantee: aggregate statistics and interaction transcripts may still leak information in principle. We therefore evaluate membership-inference-style leakage risks in Appendix E. Proposals are generated in small batches per iteration to control exploration of the combinatorial rule space.

**Deterministic validation module.** Each proposed rule is deterministically evaluated using the tool interface. The validation module enforces both statistical performance and grammar-level deployability constraints. A rule $r$ is retained only if it satisfies a minimum rule-level discrimination threshold $\tau_{\text{rule}}$ and passes redundancy checks with respect to previously retained rules:

$$\text{AUROC}(r) \geq \tau_{\text{rule}}, \qquad \max_{r' \in \mathcal{P}} J^+(r, r') \leq \delta,$$

where $\mathcal{P}$ denotes the current pool of retained rules, and $J^+$ denotes Jaccard similarity computed over positive-class coverage vectors on $\mathcal{D}_{\text{val}}$. Rules failing either criterion are discarded without further consideration.

The threshold $\tau_{\text{rule}}$ is used only as a coarse screening hyperparameter, not as a task-specific decision threshold and not as a learned parameter for individual rules. Its purpose is to remove clearly non-informative candidates while preserving enough weak but complementary rules for downstream checklist assembly. We therefore set $\tau_{\text{rule}}$ deliberately low, requiring each retained rule to have at least modest standalone discrimination while allowing the final checklist evaluation to select combinations of complementary signals.

**Clinical plausibility agent.** All rules passing deterministic validation are subsequently reviewed by a clinical plausibility agent, an independent LLM-based self-check. This agent evaluates whether a rule corresponds to a coherent and defensible clinical statement given the variable semantics and transformations used. Rules judged implausible or clinically nonsensical are rejected *prior to final retention*, ensuring that only rules satisfying both statistical and semantic criteria enter the candidate pool. Importantly, this filter is purely eliminative: it does not propose, modify, or rank rules, and cannot introduce new structure into the score.

**Score construction agent.** Given the retained pool $\mathcal{P}$, a third agent proposes a checklist $\mathcal{S} \subseteq \mathcal{P}$ with $|\mathcal{S}| \leq m$. We employ an agent for this step to bias checklist assembly toward semantically diverse rule combinations. Classical

solvers such as MIP can also be applied to the retained rule pool, and we evaluate such variants in Appendix E. In our experiments, purely objective-driven selection can favor statistically similar proxies for the same underlying signal, whereas the agent can propose checklists spanning distinct physiological domains, such as respiratory, cardiovascular, and renal signals. Each proposed checklist is evaluated deterministically via the tool interface to yield discrimination and coverage metrics. Across proposals and refinement steps, we retain the best-performing checklist on $\mathcal{D}_{\text{val}}$ under the target utility $\mathcal{U}$, ensuring that semantic guidance affects which subsets are explored while final selection remains grounded in validation performance.

**Iterative refinement.** The score construction agent may iteratively refine its proposal over a fixed number of steps, receiving updated evaluation metrics after each revision. Refinement operations are limited to rule inclusion or exclusion within the retained pool $\mathcal{P}$. Importantly, the decision threshold $\tau$ defining the positive class (i.e., $S(\tilde{\mathbf{x}}) \geq \tau$) is optimized automatically by the evaluation tools by maximizing Youden's $J$ statistic on the data and is *not* selected by the agent.

## 5. Experiments

We evaluate the clinical scoring systems learned by `AgentScore` across a diverse set of real-world clinical prediction tasks. Our evaluation is structured around four questions:

(i) **Predictive adequacy:** Can deployable, unit-weighted checklists learned by `AgentScore` achieve discrimination comparable to interpretable but less deployable machine learning models?

(ii) **Guideline competitiveness:** How do the learned scores compare to established clinical guidelines under realistic cross-institutional evaluation?

(iii) **Practical deployability:** Do the resulting scoring systems satisfy the cognitive and operational constraints required for bedside use?

(iv) **Ablation study:** Which components of the multi-step agent-based search procedure contribute to performance under these constraints?

Unless otherwise stated, all main experiments use GPT-5 as the agent backbone. Additional analyses of the learned guidelines, comparisons across different LLM backbones, privacy-leakage audits, and controlled experiments giving baseline methods access to `AgentScore`-generated features are provided in Appendix E.

### 5.1. Predictive Performance

**Datasets and Tasks.** We evaluate `AgentScore` on eight real-world clinical prediction tasks derived from the publicly available MIMIC-IV and eICU EHR datasets, spanning mortality prediction, length-of-stay prediction, and prediction of clinically actionable interventions. All tasks use only variables routinely available at the time of clinical decision-making. We perform 5-fold cross-validation, holding out 20% of patients in each fold as a test set used exclusively for final evaluation. Reported metrics are computed on held-out test splits and aggregated across folds. For `AgentScore`, we further split the training portion, using 20% of the training patients as validation. Additional details are provided in Appendix D.

**Baselines.** We compare `AgentScore` against a comprehensive set of interpretable and score-learning approaches representing strong alternatives for clinical risk prediction under different deployability constraints. These include state-of-the-art integer-valued and checklist scoring methods, including RiskSLIM, FasterRisk, and Optimal Checklists, as well as pooled penalized logistic regression (PLR) baselines (Liu et al., 2022). RiskSLIM and FasterRisk learn sparse integer-weighted scores, while Optimal Checklists learns unit-weighted $N$-of-$M$ checklists over fixed candidate features. These methods provide strong score-learning baselines, but they either rely on non-unit coefficients or assume a pre-specified feature space, and therefore do not jointly construct and select clinically meaningful derived rules under checklist deployability constraints. As less constrained interpretable comparators, we additionally evaluate logistic regression, decision trees, and the AutoScore framework. These models can achieve strong discrimination but rely on flexible coefficients, post-hoc discretization, or model-execution pipelines that do not satisfy the same guideline-style checklist constraints.

**Results.** Among score-learning baselines, `AgentScore` improves over integer-weighted methods such as RiskSLIM and FasterRisk, as well as over the unit-weighted Optimal Checklists baseline, despite operating under the additional requirement that clinically meaningful derived rules are constructed rather than supplied as fixed candidate features or arbitrary value bins. We include pooled PLR baselines primarily to illustrate the brittleness of post-hoc coefficient modification in logistic models. Logistic regression, decision trees, and AutoScore can achieve higher AUROC in some settings, but they are not checklist-deployable by design: they rely on learned non-unit coefficients, post-hoc discretization, or execution requirements that impose arithmetic and operational overhead relative to unit-weighted clinical checklists.

Averaged across all eight tasks, `AgentScore` significantly improves AUROC over integer score-learning baselines,

*Table 3.* **Model performances.** `AgentScore` enforces unit weights, bounded size, checklist structure, and automatic derived-rule construction. Competing methods either operate over fixed feature encodings or relax one or more checklist-deployability constraints. Entries are mean ± std. For each dataset, the best-performing score-based method (highest mean AUROC) is shown in **bold**.

| Method | MIMIC AF | MIMIC COPD | MIMIC HF | MIMIC AKI | MIMIC Cancer | MIMIC Lung | eICU LOS | eICU Vaso | Mean |
|---|---|---|---|---|---|---|---|---|---|
| Decision Tree | 0.79 ± 0.01 | 0.69 ± 0.01 | 0.77 ± 0.02 | 0.85 ± 0.00 | 0.64 ± 0.01 | 0.66 ± 0.00 | 0.69 ± 0.00 | 0.77 ± 0.00 | 0.73 ± 0.07 |
| Logistic Regression | 0.77 ± 0.01 | 0.68 ± 0.01 | 0.76 ± 0.01 | 0.78 ± 0.00 | 0.64 ± 0.02 | 0.65 ± 0.01 | 0.69 ± 0.00 | 0.75 ± 0.00 | 0.72 ± 0.05 |
| AutoScore | 0.82 ± 0.00 | 0.66 ± 0.01 | 0.81 ± 0.01 | 0.84 ± 0.00 | 0.60 ± 0.01 | 0.66 ± 0.00 | 0.63 ± 0.00 | 0.75 ± 0.00 | 0.72 ± 0.09 |
| FasterRisk | 0.75 ± 0.02 | 0.62 ± 0.00 | 0.73 ± 0.01 | 0.76 ± 0.00 | 0.57 ± 0.01 | 0.54 ± 0.00 | **0.70 ± 0.00** | 0.75 ± 0.00 | 0.68 ± 0.08 |
| RiskSLIM | 0.75 ± 0.01 | 0.62 ± 0.00 | 0.74 ± 0.01 | **0.80 ± 0.00** | 0.58 ± 0.01 | 0.56 ± 0.03 | 0.62 ± 0.00 | 0.64 ± 0.01 | 0.66 ± 0.08 |
| Pooled PLR (RDU) | 0.70 ± 0.01 | 0.58 ± 0.00 | 0.73 ± 0.03 | 0.73 ± 0.00 | 0.55 ± 0.01 | 0.61 ± 0.01 | 0.67 ± 0.00 | 0.64 ± 0.00 | 0.65 ± 0.06 |
| Pooled PLR (RSRD) | 0.70 ± 0.01 | 0.58 ± 0.00 | 0.70 ± 0.01 | 0.76 ± 0.00 | 0.55 ± 0.01 | 0.59 ± 0.02 | 0.62 ± 0.00 | 0.64 ± 0.00 | 0.64 ± 0.07 |
| Optimal Checklists | 0.67 ± 0.06 | 0.66 ± 0.02 | 0.56 ± 0.03 | 0.68 ± 0.00 | **0.62 ± 0.02** | **0.67 ± 0.01** | 0.59 ± 0.00 | 0.54 ± 0.01 | 0.62 ± 0.05 |
| AgentScore | **0.81 ± 0.01** | 0.63 ± 0.03 | **0.79 ± 0.01** | 0.79 ± 0.02 | 0.59 ± 0.01 | 0.63 ± 0.00 | 0.67 ± 0.00 | **0.76 ± 0.00** | **0.71 ± 0.08** |

with mean fold-level gains of +0.05 versus RiskSLIM and +0.03 versus FasterRisk. Two-sided paired tests on fold-level AUROCs ($n = 40$; Holm–Bonferroni corrected) reject equality for all score-based baselines ($p < 0.001$). Full results are included in Appendix E.

## 5.2. Guideline Competitiveness

In the previous section, we compared `AgentScore` against strong interpretable machine-learning baselines that do not satisfy guideline-style deployability constraints. We now evaluate a different and clinically relevant question: whether data-driven, deployable scoring systems learned by `AgentScore` can compete with *established clinical guidelines* under out-of-distribution evaluation settings that mimic the validation procedures used prior to clinical deployment.

**Evaluation protocol.** We evaluate guideline competitiveness under a cross-institutional setting. `AgentScore` is trained on data from one institution and evaluated on an independent external cohort. For fairness, existing clinical guidelines are applied without modification to their rule structure; only the decision threshold is calibrated on the same training data used for `AgentScore`, isolating the quality of the underlying rule sets rather than threshold selection.

**Results.** Using a PhysioNet ICU 2012 mortality benchmark, we compare `AgentScore` against the SOFA (Vincent et al., 1996) and SAPS-I scores (Gall et al., 1984), two widely used ICU risk stratification tools. To further assess robustness across healthcare systems and disease domains, we train `AgentScore` on a UK cystic fibrosis cohort and evaluate performance on an independent Canadian cohort. We compare against commonly used lung-transplantation eligibility guidelines for identifying high-risk individuals (Ramos et al., 2019) and a simple, widely used $FEV_1$-based threshold ($FEV_1 < 30\%$) (Ramos et al., 2017). Figure 3 shows that `AgentScore` achieves higher discrimination than the established guidelines under external validation. These results suggest that data-driven checklist construction can recover externally portable risk rules that remain operationally comparable to existing clinical guidelines.

## 5.3. Practical Deployability

We conducted a structured expert review of score deployability with a panel of $N = 18$ practicing clinicians (89% with $\geq 6$ years of clinical experience) from six countries. Participants evaluated four representative `AgentScore` checklists alongside matched outputs from FasterRisk; they were instructed to assume equal predictive performance. Across 72 pairwise judgments per question, participants significantly preferred `AgentScore`-generated checklists for alignment with guideline-style reasoning (85% vs. 15%; binomial $p < 10^{-9}$, Cohen's $h = 0.77$), ease of bedside application (71% vs. 29%; $p < 10^{-3}$, $h = 0.43$), and deployment preference (81% vs. 19%; $p < 10^{-7}$, $h = 0.66$). For overall model preference, 67% of clinicians selected `AgentScore`, 6% selected `FasterRisk`, and 28% reported no preference ($\chi^2(2) = 10.3$, $p = 0.006$). Full methodology, question wording, and per-question response distributions are provided in Appendix E.9.

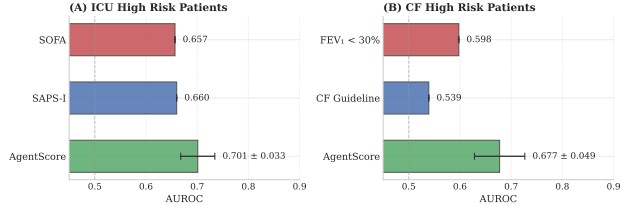

*Figure 3.* **External validation against guidelines.** `AgentScore` achieves higher AUROC than the clinical guidelines in both external-validation settings. (mean ± std over 5 seeds; guidelines deterministic).

## 5.4. Ablation Studies

We conduct targeted ablation studies to isolate the contribution of key components of `AgentScore` (see Table 4). In the *LLM Only* ablation, we replace the constrained proposal–evaluation loop with a single unconstrained language model, removing tool-mediated validation and iterative feedback. In the *Single-Pass* ablation, we disable iterative refinement while retaining the evaluation tools. We further ablate structural constraints. The *No Jaccard* variant disables redundancy filtering based on positive-class Jaccard overlap, allowing highly overlapping rules to co-exist. The *No Diversity* variant disables rule-type diversity enforcement that encourages diverse rule family sampling.

**Results.** Across all comparisons, the full `AgentScore` pipeline significantly outperforms the ablated variants under paired Wilcoxon signed-rank tests with Holm correction ($p_{\mathrm{Holm}} < 0.05$). Effect sizes are substantial: Cohen's $d$ ranges from approximately 1.0 for milder ablations to approximately 2.8 when deterministic validation is removed. These results indicate that each component contributes meaningfully to performance, with the largest degradation arising from replacing the constrained proposal–validation loop with unconstrained LLM generation.

*Table 4.* **Ablation study of `AgentScore` components.** Mean AUROC is averaged across datasets, with standard deviation computed across tasks.

| Ablation | Mean AUROC | Std |
|---|---|---|
| Full `AgentScore` | **0.71** | 0.08 |
| LLM Only (unconstrained) | 0.59 | 0.07 |
| Single-Pass (no refinement) | 0.69 | 0.07 |
| No Jaccard | 0.69 | 0.08 |
| No Rule Diversity | 0.70 | 0.08 |

## 6. Discussion

Despite high predictive accuracy in retrospective evaluations, many clinical machine-learning models fail to translate into real-world impact. A central reason is persistent misalignment with clinical workflows: complex models often depend on dense or non-routinely available inputs, specialized computational infrastructure, and opaque inference procedures. These barriers are particularly pronounced in resource-constrained settings. In this work, we argue that meaningful clinical impact does not always require increasingly large or expressive models. Instead, it requires leveraging machine learning to *improve existing clinical workflows* while respecting the structural, cognitive, and operational constraints under which medical decisions are made. From this perspective, clinical scoring systems are not a legacy artifact to be replaced, but a deliberately constrained hypothesis class optimized for deployment, where strong performance depends not only on selecting a sparse checklist, but also on constructing clinically meaningful rules from raw variables.

We introduce `AgentScore`, a constrained learning framework that automatically constructs unit-weighted clinical checklists under explicit deployability constraints. Computational resources are required only during development; deployment reduces to evaluating a small set of binary rules and summing their outputs, enabling bedside use without calculators, servers, or continuous model maintenance. Beyond deployability, `AgentScore` addresses a key barrier to clinical ML adoption: data governance. Rule generation is mediated through a restricted tool interface and decoupled from direct data access. Large language models never observe raw patient records and interact only through a restricted tool interface that returns aggregate statistics. This can simplify governance and audit in collaborations where direct access to patient-level data is limited, although aggregate interfaces still require careful privacy controls.

More broadly, our results suggest that much of the predictive signal exploited by flexible interpretable models can be retained within a tightly constrained, deployable hypothesis class when semantic rule construction, deterministic evaluation, and structured selection are jointly enforced. This challenges a common implicit assumption that clinical machine learning must trade deployability for performance, and highlights constrained optimization over guideline-compatible model classes as a promising direction for future research.

**Limitations.** Clinical scoring systems necessarily trade expressivity for simplicity. While such systems can standardize care and improve outcomes at the population level, they cannot capture all nuances of individual patient trajectories, and clinical judgment remains essential. Since `AgentScore` is less expressive than flexible statistical or black-box models, it is not expected to consistently outperform them in raw predictive performance. Rule discovery is also *knowledge-bounded*: although acceptance is data-grounded via deterministic evaluation, the candidate rules are mediated by the LLM and limited by the constructs it can surface; thus, predictive relationships may be missed if they are not semantically accessible to the agent. Furthermore, `AgentScore` prioritizes semantic meaningfulness and deployability over formal optimization guarantees and does not guarantee convergence to a globally optimal checklist over the full derived-rule space. While our framework enforces deployability constraints, it does not explicitly enforce fairness constraints across demographic subgroups. Like all data-driven methods, `AgentScore` may learn rules that reflect historical biases in clinical practice or documentation patterns; formal fairness auditing and subgroup validation remain essential prior to deployment. Finally, while our framework reduces reliance on direct data access during learning, it does not eliminate the need for careful dataset curation and does not by itself guarantee robustness under distribution shift.

**Conclusion and clinical perspective.** `AgentScore` demonstrates that machine learning can generate candidate scoring systems aligned with guideline-style use and suitable for prospective validation, rather than replacing clinical guidelines. By learning compact, interpretable, and auditable unit-weighted checklists from clinically meaningful derived rules, our approach provides a pathway toward deployable-by-design clinical ML. More broadly, `AgentScore` exemplifies constrained semantic optimization: a framework in which hypothesis class design, deployability, and verification are treated as first-class modeling concerns.

## Acknowledgments

We thank the anonymous ICML reviewers for their comments and suggestions. This work was supported by Azure sponsorship credits granted by Microsoft's AI for Good Research Lab. C.C. gratefully acknowledges funding from Apple. The Cambridge Centre for AI in Medicine (CCAIM) receives funding from GSK, Boehringer-Ingelheim, AstraZeneca, Sanofi and Quantum Black, AI by McKinsey. Figures 1 and 2 were created using BioRender.

## Impact statement

This paper presents work whose goal is to advance the field of machine learning by proposing a framework for constructing interpretable checklist models under explicit deployability constraints. The scoring systems produced in this paper are research artifacts only and are not intended for clinical use. As with any model trained on observational data, they may reflect bias or dataset artifacts and could cause harm if deployed without expert review, external validation, and regulatory oversight. Our aim is to contribute a methodological perspective on clinically aligned model design, not to promote real-world deployment. We release the code for `AgentScore` under `https://github.com/Sr933/agentscore-official` and `https://github.com/vanderschaarlab/agentscore-official`.

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

# Supplementary Material for `AgentScore`

## A. Clinical scoring systems

### A.1. Why clinical scoring systems matter

**Role in evidence-based medicine and workflow standardization.** In routine clinical practice, guidelines function as operational coordination mechanisms that determine how evidence is applied across heterogeneous clinicians, settings, and time horizons, often under conditions of uncertainty and time pressure (Welch et al., 2022; Dimmer et al., 2024). Their effectiveness depends not only on statistical validity, but on whether recommendations can be executed reliably at the bedside using routinely available information, frequently by clinicians with differing levels of training (Graham et al., 2011). As a result, guideline impact in real-world workflows is driven less by marginal improvements in predictive accuracy than by the ability to produce decisions that are consistent, auditable, and directly actionable (Zhong et al., 2025).

Clinical scoring systems instantiate these principles by mapping a small number of routinely collected patient features to discrete risk strata or management recommendations. By coupling bounded integer scores to explicit action thresholds (e.g., hospital admission, advanced imaging, antibiotic administration), they provide a shared decision representation that supports triage, escalation, and communication across care settings (Dambha-Miller et al., 2020; Olesen et al., 2012; Vincent & Moreno, 2010). Crucially, these scores are embedded directly into care pathways rather than interpreted as unconstrained predictive outputs.

**Why small, explicit rules survive deployment.** From a learning-theoretic perspective, clinical deployment induces a mismatch between hypothesis classes that are easy to optimize and those that are viable in practice (Wang, 2025). High-capacity models benefit from smooth parameterizations and weak structural constraints, enabling efficient optimization and strong retrospective performance. In contrast, bedside deployment restricts models to discrete, low-capacity, and tightly structured forms that must be executed reliably under time pressure and partial information (Graham et al., 2011).

Under these constraints, additional model flexibility often yields diminishing returns with respect to clinically actionable decision quality (Chen, 2020; Steyerberg et al., 2010). Instead, compact rule-based models operate near the boundary of minimal sufficient complexity: by excluding degrees of freedom that cannot be supported at deployment time, they exhibit improved robustness to sampling variability, distribution shift, and execution noise (Dash & Liu, 1997; Wasylewicz & Scheepers-Hoeks, 2018). Their enduring use in clinical practice reflects an implicit optimization objective: maximize performance subject to hard constraints on executability and stability, rather than unconstrained predictive accuracy.

### A.2. Case studies of clinical impact

Clinical scoring systems achieve impact not merely by ranking risk, but by reliably changing clinical decisions through explicit, thresholded actions. Table 5 summarizes the checklist structure of four widely adopted decision rules that illustrate complementary mechanisms of clinical impact. Across all cases, clinical impact arises from the presence of trusted, explicit action thresholds that deterministically link observed findings to management decisions. Their continued use reflects the clinical value of compact, unit-weighted checklists: they translate uncertainty into reliable action, rather than merely maximizing discriminative accuracy.

Acute care escalation tools such as qSOFA (Singer et al., 2016) and CURB-65 (Lim et al., 2003) provide rapid bedside identification of patients at elevated risk of deterioration or mortality. By mapping a small number of routinely available observations to discrete severity thresholds, these scores support timely escalation, admission decisions, and resource allocation under conditions of uncertainty. In primary care, the Centor criteria illustrate a complementary mechanism: by aggregating a small set of binary clinical findings into a bounded checklist, the score supports rational antibiotic prescribing and targeted diagnostic testing, reducing unnecessary treatment while maintaining safety. The Ottawa Ankle Rules (Shell, 1993) demonstrate a distinct but equally important form of impact in emergency medicine, where a conservative binary checklist enables the safe exclusion of fracture and avoids unnecessary imaging, reducing cost and patient burden without increasing missed injuries. For the Ottawa Ankle Rules, the checklist is operationalized as a conservative OR-rule.

*Table 5.* **Binary checklist structure of representative clinical decision rules.** Each system consists of unit-weighted binary conditions (+1 if satisfied) whose total score is mapped to an explicit guideline-recommended clinical action.

| Score | Item | Binary rule (satisfied?) | Points |
|---|---|---|---|
| **qSOFA** | Respiratory rate | Respiratory rate $\geq 22$ breaths/min | +1 |
| | Blood pressure | Systolic blood pressure $\leq 100$ mmHg | +1 |
| | Mental status | Altered mental status (GCS $< 15$) | +1 |
| | *Threshold & action* | Score $\geq 2 \Rightarrow$ high risk of sepsis; urgent evaluation | |
| **CURB-65** | Confusion | New onset confusion (disorientation to person, place, or time) | +1 |
| | Urea | Blood urea nitrogen $> 7$ mmol/L | +1 |
| | Respiratory rate | Respiratory rate $\geq 30$ breaths/min | +1 |
| | Blood pressure | Systolic $< 90$ mmHg or diastolic $\leq 60$ mmHg | +1 |
| | Age | Age $\geq 65$ years | +1 |
| | *Threshold & action* | Score $\geq 3 \Rightarrow$ inpatient management recommended | |
| **Centor** | Fever | History of fever or measured temperature $\geq 38°$C | +1 |
| | Tonsillar exudates | Tonsillar exudates or swelling present | +1 |
| | Cervical lymphadenopathy | Tender anterior cervical lymph nodes | +1 |
| | Cough absence | Absence of cough | +1 |
| | *Threshold & action* | Score $\geq 3 \Rightarrow$ consider antibiotics or rapid strep testing | |
| **Ottawa Ankle Rules** | Malleolar pain | Pain in the malleolar zone | +1 |
| | Bone tenderness (lateral) | Tenderness at posterior edge or tip of lateral malleolus | +1 |
| | Bone tenderness (medial) | Tenderness at posterior edge or tip of medial malleolus | +1 |
| | Weight bearing (immediate) | Inability to bear weight immediately after injury | +1 |
| | Weight bearing (ED) | Inability to bear weight for four steps in the hospital | +1 |
| | *Threshold & action* | Score $\geq 1 \Rightarrow$ ankle radiography indicated | |

### A.3. Survey of established guideline scores

**Scope.** To contextualize our modeling choices, we examined a range of widely adopted clinical scoring systems spanning acute care, cardiology, respiratory disease, obstetrics, psychiatry, and trauma (Table 6). We focused on scores that are guideline-embedded, widely cited, and routinely used in clinical practice. Despite their diverse clinical contexts and historical origins, these systems exhibit a remarkably consistent design philosophy, suggesting the presence of shared, domain-agnostic deployment constraints.

**Common structural properties.** Across domains, successful clinical scoring systems share several recurring features: (i) a small number of easily recalled items (typically 5–10); (ii) reliance on routine, low-cost inputs; (iii) integer or low-cardinality scoring that enables manual computation; (iv) bounded score ranges that support clear risk stratification; and (v) explicit thresholds tied to concrete management actions. Together, these properties promote robustness to missingness, reduce cognitive burden, and enable reliable execution without specialized infrastructure.

**Implications for hypothesis class design.** These empirical regularities motivate learning directly within the checklist structures observed in deployed guidelines, rather than approximating them post hoc. In particular, across surveyed systems the median checklist length is five, with the majority of widely deployed scores operating at six or fewer items (Table 6). This concentration at small rule counts supports treating a limited rule budget as a deployability prior rather than a tunable performance parameter. Restricting the rule budget regularizes the hypothesis class, limits cognitive load, and aligns model structure with real-world guideline practice. This is also consistent with classic results on the limits of human working memory, suggesting that compact checklists better match the cognitive constraints of bedside decision-making (Miller, 1956).

Similarly, restricting attention to unit-weighted rules is supported by both clinical convention and classical results on improper linear models, which show that equal-weighted predictors often perform competitively in noisy, low-signal regimes while exhibiting greater robustness to estimation error (Dawes, 1979). While some successful scores employ non-unit weights, their increased arithmetic complexity frequently necessitates reference aids. When a unit-weighted formulation is feasible, it is therefore preferable due to reduced arithmetic burden and more reliable bedside execution.

Taken together, these considerations justify focusing on compact, unit-weighted checklists as the most expressive hypothesis class that remains reliably deployable under realistic clinical conditions.

*Table 6.* **Overview of established clinical scoring systems.** Widely adopted scores rely on a small number of routine features, integer or categorical point allocations, and explicit decision thresholds, enabling memorability and bedside deployment without specialized infrastructure.

| Domain | Score | # Items | Range | Typical use / threshold | Reference |
|---|---|---|---|---|---|
| Infection / Sepsis | SIRS | 4 | 0–4 | $\geq$ 2: systemic inflammation | (Bone et al., 1992) |
| Thrombosis | Wells (DVT) | 9 | 0–9 | $\geq$ 2: high DVT probability | (Wells et al., 1997) |
| | Wells (PE) | 7 | 0–12.5 | $\geq$ 4: high PE probability | (Wells et al., 2000) |
| Cardiology | CHA$_2$DS$_2$-VASc | 8 | 0–9 | $\geq$ 2: anticoagulation | (Lip et al., 2010) |
| | GRACE | 8 | 0–15 | ACS mortality risk | (Granger, 2003) |
| | Framingham | 9 | 0–30 | 10-year cardiovascular risk | (D'Agostino et al., 2008) |
| Respiratory | CURB-65 | 5 | 0–5 | $\geq$ 3: inpatient management | (Lim et al., 2003) |
| | Centor | 4 | 0–4 | $\geq$ 3: consider antibiotics | (Centor et al., 1981) |
| Gastrointestinal | Glasgow–Blatchford | 8 | 0–23 | $\geq$ 1: urgent endoscopy | (Blatchford et al., 2000) |
| Neurology | Glasgow Coma Scale | 3 | 3–15 | $\leq$ 8: severe head injury | (Teasdale & Jennett, 1974) |
| Psychiatry | PHQ-9 | 9 | 0–27 | $\geq$ 10: moderate depression | (Kroenke et al., 2001) |
| Obstetrics / Neonatal | Bishop Score | 5 | 0–13 | $\geq$ 8: induction readiness | (Bishop, 1964) |
| | Silverman–Andersen | 5 | 0–10 | $\geq$ 7: respiratory distress | (Silverman & Andersen, 1956) |
| Critical Care | SOFA | 6 | 0–24 | $\geq$ 2: organ dysfunction | (Vincent et al., 1996) |
| | NEWS2 | 6 | 0–20 | $\geq$ 5: urgent clinical review | (Royal College of Physicians, 2017) |
| | qSOFA | 3 | 0–3 | $\geq$ 2: sepsis risk trigger | (Singer et al., 2016) |
| Trauma / Imaging | RTS | 4 | 0–8 | $\geq$ 2: trauma team activation | (Champion et al., 1989) |
| | Ottawa Ankle Rules | 5 | 0–5 | $\geq$ 1: ankle radiography | (Shell, 1993) |

## B. The Clinical Scoring System Rule Grammar

Our rule language is designed to capture the *forms of reasoning that already survive deployment* in clinical guidelines: rules must be actionable (paired with clear thresholds), auditable (each condition can be inspected and contested), compute-free at the bedside (no continuous inference pipeline), and compatible with routine documentation and measurement practices (Woolf et al., 1999; Graham et al., 2011; Shortliffe & Sepulveda, 2018; Rudin, 2019; Moons et al., 2015). Accordingly, we restrict attention to a small set of rule families that recur across widely adopted scoring systems.

**(i) Numeric thresholds: cutpoints as operational triggers.** Clinical guidelines frequently operationalize physiologic risk through thresholded cutpoints that map directly to escalation actions. For example, CURB-65 uses a urea cutpoint (blood urea nitrogen $> 7$ mmol/L) and a respiratory rate cutpoint ($\geq 30$ breaths/min) to define higher-severity pneumonia and guide inpatient management (Lim et al., 2003). Similarly, qSOFA uses binary thresholds (e.g., respiratory rate $\geq 22$/min; systolic blood pressure $\leq 100$ mmHg) as a sepsis risk trigger (Singer et al., 2016). In practice, such thresholds are robust to moderate measurement noise and are easy to execute at the bedside, motivating the inclusion of single-variable threshold rules as a primitive in $\mathcal{R}$.

**(ii) Numeric ranges: normality bands and safety windows.** Many physiologic variables exhibit *both* low- and high-risk regimes, and guidelines often reason in terms of safe bands or abnormality windows rather than monotone risk. Early warning scores, including NEWS2, discretize vital signs into clinically meaningful ranges (e.g., oxygen saturation bands, temperature bands, and systolic blood pressure bands) that reflect nonlinear risk and guide escalation (Royal College of Physicians, 2017). Range rules therefore provide a compact mechanism to encode "normal vs abnormal" intervals without introducing nontransparent nonlinearities.

**(iii) Categorical inclusion and (iv) binary presence: yes/no clinical facts.** Guideline logic often depends on discrete clinical facts, such as diagnoses, comorbidities, and prior events, that are naturally represented as categorical inclusion or binary presence rules. For instance, CHA$_2$DS$_2$-VASc assigns points for discrete risk factors such as diabetes mellitus,

prior stroke/TIA/thromboembolism, vascular disease, and heart failure (Lip et al., 2010; Olesen et al., 2012). Similarly, the Wells criteria include categorical items such as active cancer and recent immobilization/surgery (Wells et al., 1997). These variables are typically well documented in routine care and map cleanly to checklist items, motivating explicit support for categorical and binary predicates.

**(v) Ratios and contrasts: derived physiology with long precedent.** Clinicians frequently reason in *derived indices* that normalize or compare measurements rather than relying on raw values alone. Although many guideline scores encode derived reasoning implicitly via multiple thresholded items, the underlying clinical logic often corresponds to ratios or contrasts (e.g., oxygenation indices such as $PaO_2/FiO_2$, perfusion indices, or relative changes between related labs). Our language therefore permits a restricted allowlist of simple transformations (ratios and differences) to capture such reasoning while preserving interpretability and auditability. Importantly, these derived quantities are typically available at no additional cognitive or computational cost, as they can be precomputed automatically when the relevant laboratory measurements are ordered and results returned. We do not allow arbitrary algebraic expressions: only clinically interpretable forms (e.g., $x_a/x_b$ or $x_a - x_b$) are permitted, and all derived rules are validated empirically and screened for plausibility by the eliminative plausibility agent.

**(vi) Count-based rules: syndromic criteria and cumulative burden.** Many clinical definitions and escalation triggers are inherently $M$-*of*-$N$ criteria: the presence of multiple abnormal findings jointly increases suspicion or defines a syndrome. Classic examples include SIRS-style criteria, which combine several abnormal vitals/labs, and organ dysfunction frameworks such as SOFA, which aggregate multiple organ-specific components (Bone et al., 1992; Vincent et al., 1996). A widely taught diagnostic example is the Duke criteria for infective endocarditis, which require combinations of major and minor criteria to establish or exclude diagnosis, illustrating how guideline reasoning often formalizes evidence accumulation as transparent count-based logic (Durack et al., 1994). Count-based rules preserve transparency because the contributing items remain explicit and auditable; they also align naturally with unit-weighted checklist execution by capturing cumulative burden without complex arithmetic.

**(vii) Shallow logical composition: small conjunctions/disjunctions mirror guideline logic.** Guidelines often specify short conjunctions and disjunctions that define an abnormality or escalation trigger. For example, CURB-65 assigns a point for hypotension defined as *systolic* blood pressure $< 90$ **or** *diastolic* $\leq 60$ mmHg (Lim et al., 2003), and the Wells criteria include a competing-diagnosis clause ("alternative diagnosis at least as likely") that explicitly modifies the risk assessment (Wells et al., 1997). We therefore allow shallow AND/OR compositions over base predicates while restricting logical depth to preserve memorability and reduce execution error under time pressure (Miller, 1956; Gigerenzer & Gaissmaier, 2011).

**(viii) Temporal and distributional summaries: capturing acuity and trend without black-box time-series.** Clinical deterioration is often defined by *trends* rather than static values (e.g., rapid worsening, sustained abnormalities, or recent changes). In practice, many guideline tools operationalize time indirectly through repeated observations and escalation thresholds (e.g., repeated NEWS2 scoring over time) (Royal College of Physicians, 2017). To capture such reasoning while remaining compute-free at deployment, we restrict temporal constructs to simple, auditable summaries over a pre-index window (e.g., delta, percent change, range, or max–min). Canonical examples include acute kidney injury (AKI), where diagnostic criteria depend on changes in serum creatinine over time (e.g., a rise from baseline within a short window) rather than a single absolute measurement (Bellomo et al., 2004). Similar trend-based reasoning appears in myocardial infarction evaluation, where serial troponin measurements are interpreted through rises and/or falls over time, and in sepsis management, where lactate clearance (a decrease in lactate over a specified interval) is used to assess response to resuscitation and ongoing risk. More broadly, clinicians routinely monitor temporal patterns such as the fever curve (persistence or trajectory of temperature) when assessing infection progression or treatment response. All temporal features are computed using only measurements available prior to the prediction time to avoid label leakage.

**Clinically meaningful discretization.** To avoid spurious cutpoints and improve interpretability, continuous thresholds are quantized to clinically meaningful values (e.g., integer vitals, common lab units, or guideline-style bands), consistent with classical regression-to-points approaches used in clinical score construction (Sullivan et al., 2004). This discretization stabilizes rules under sampling variability and supports reliable bedside execution.

**Missingness and measurement frequency.** EHR data are characterized by irregular sampling and informative missingness. To ensure deployability and avoid hidden imputation effects, we define explicit missingness semantics for each rule family and report them alongside learned checklists. Concretely, if a required input is missing at prediction time and cannot be imputed reliably, the rule evaluates to false by default, preserving conservative, audit-friendly behavior under missingness.

# C. Extended Related Work

## C.1. Clinical scoring systems and score construction

**Expert-designed scoring systems.** A large fraction of widely used clinical scores are developed through expert consensus and guideline processes rather than direct optimization on large datasets. Such scores typically encode domain knowledge and pragmatic workflow constraints (e.g., reliance on routinely available measurements, bounded item counts, and actionable thresholds) and thus benefit from strong face validity, institutional trust, and straightforward bedside execution. In evidence-based medicine, these properties are not incidental: guidelines function as coordination mechanisms that standardize decisions across heterogeneous clinicians and settings, and therefore prioritize auditability and actionability alongside predictive utility (Woolf et al., 1999). The limitations are equally well known: expert-derived scores can be conservative, slow to update as evidence evolves, and may underperform in new populations or under distribution shift, especially when derived from small cohorts or historical practice patterns (Challener et al., 2019). These concerns are now formalized in clinical prediction model methodology through reporting and bias-assessment frameworks such as TRIPOD and PROBAST, which emphasize transparency, applicability, and robustness under dataset shift (Collins et al., 2015; Wolff et al., 2019).

**Regression-based score derivation and rounding.** A classical approach to score construction fits a regression model (often logistic regression) and converts continuous coefficients into an integer point system via scaling and rounding, as exemplified by Framingham-style risk scores (Sullivan et al., 2004). This pipeline preserves the additive structure of linear predictors while producing a human-usable point tally, and it is often accompanied by calibration to produce risk estimates. However, coefficient-to-point conversion introduces additional approximation error, and the resulting point weights are typically non-unit and heterogeneous, imposing arithmetic burden and reducing memorability. Moreover, estimated coefficients may be unstable under sampling variability, collinearity, and missingness mechanisms, leading to sensitivity in the derived point system and thresholds across cohorts (Sullivan et al., 2004; Moons et al., 2015). These issues are particularly salient when scores are intended for manual computation or when guideline governance requires stable, transparent rules that remain sensible across institutions.

**Optimization-based integer scoring methods.** Recent work has formalized score construction as an optimization problem over sparse integer-weighted linear models, enabling direct control over sparsity, coefficient bounds, and (in some cases) monotonicity or sign constraints. Representative examples include AutoScore, which combines feature ranking and discretization with scorecard-style point assignment; RiskSLIM, which solves a mixed-integer program to optimize a sparse integer score; and FasterRisk, which provides a scalable approximation procedure for learning sparse risk scores under coefficient and sparsity constraints (Xie et al., 2020; Ustun & Rudin, 2015; Liu et al., 2022). These methods offer substantial advantages over ad hoc rounding: they search over discrete coefficient spaces directly and provide clearer objective-driven trade-offs. Nonetheless, they remain fundamentally *selectors* over a pre-specified feature matrix; in practice, this requires committing to a fixed set of binned threshold indicators and does not address the *generation-selection* gap when the clinically meaningful hypothesis class includes derived quantities, temporal summaries, or shallow compositional logic that are not explicitly available as features. Further, even when coefficients are small integers, heterogeneous weights typically require nontrivial arithmetic at the bedside and do not inherently yield checklist-style $N$-of-$M$ decision procedures.

**Why integer coefficients alone are insufficient.** While integer coefficients increase interpretability relative to continuous weights, *integer* does not imply *deployable-by-design*. These human-factors limitations are increasingly recognized in clinical AI evaluation guidelines, which stress that executable decision procedures must be assessed in context of use rather than solely via retrospective accuracy metrics (Moons et al., 2025; Kwong et al., 2022). Heterogeneous point values still impose cognitive load and increase the risk of arithmetic errors under interruption and time pressure, and they often require external aids (charts, calculators, or EHR integration) for reliable execution. *There are, however, settings where heterogeneous integer weights are entirely appropriate:* for example, in longitudinal risk stratification and care planning where decisions are made with more time, or when scores are routinely pre-computed by software (e.g., EHR dashboards or clinical decision support systems), so the human does not perform arithmetic at the point of care. In these contexts, the added flexibility of non-unit integer weights can improve calibration or risk separation without materially increasing workflow burden. The central issue is therefore not whether weights are integer, but whether the computation and decision procedure are *bedside-executable* under realistic operating conditions.

In contrast, unit-weighted checklists align with how clinicians frequently reason in practice: as short sets of explicit triggers where the contribution of each rule is identical and the decision reduces to a count exceeding a threshold. This design

is also supported by classical results on so-called improper linear models, which show that equal-weighted predictors can be competitive in noisy regimes and may be more robust to estimation error than finely tuned weights (Dawes, 1979; Zhang et al., 2021). From a human-factors standpoint, bounded item counts and simple aggregation rules (including $N$-of-$M$ procedures) help accommodate working-memory constraints and facilitate consistent application in high-stakes environments (Miller, 1956; Gigerenzer & Gaissmaier, 2011). Finally, integer-score learners typically inherit the expressivity limits of their pre-binning: if clinically meaningful constructs are absent from the initial feature representation, coefficient optimization alone cannot recover them.

### C.2. Interpretable machine learning

**Inherently interpretable models.** Inherently interpretable models aim to make the full predictive mechanism transparent, commonly including generalized linear models, decision trees, sparse linear models, and rule-based classifiers. Logistic regression and linear models offer additive structure and straightforward coefficient interpretation; shallow decision trees provide explicit decision paths that can be inspected and audited. More specialized rule learners include Bayesian Rule Lists (Letham et al., 2015) and constraint-based rule sets (e.g., CORELS (Angelino et al., 2018)), which produce compact if–then structures with probabilistic or combinatorial learning procedures. These approaches provide important baselines for interpretability and can be audited post hoc, but they often remain misaligned with bedside execution requirements: even sparse models may depend on dense feature sets, require preprocessing pipelines, or produce real-valued computations that are not manually executable. Moreover, many interpretable ML methods optimize within model classes (linear scores, trees, rule lists) that do not directly correspond to guideline-style checklist workflows, and thus may improve transparency without delivering *deployability* (Rudin, 2019; Molnar et al., 2022).

**Post-hoc explanations and their limitations.** Similarly, post-hoc explanation methods (e.g., SHAP, LIME) can provide useful diagnostic insight into individual predictions, but they do not constrain model behavior or guarantee stability under distribution shift (Ribeiro et al., 2016; Lundberg & Lee, 2017; Adebayo et al., 2018). More fundamentally, post-hoc explanations can be faithful only locally, can vary across equivalent representations, and may give a false sense of transparency when the underlying decision boundary remains complex (Lipton, 2018; Rudin, 2019). For guideline deployment, the central requirement is not merely that a prediction can be explained, but that the decision procedure is itself simple, auditable, and reliably executable under real-world constraints.

**Symbolic regression and rule learning.** Symbolic regression and equation discovery methods learn structured mathematical expressions from data, including frameworks such as SINDy (Brunton et al., 2016) and modern genetic-programming systems (e.g., PySR) (Cranmer, 2023). These methods aim to recover compact functional forms with scientific interpretability and have been successful in low-dimensional dynamical systems discovery. However, their objectives and constraints typically differ from clinical score design: they target continuous functional relationships rather than discrete, actionable decision rules, and they can exhibit instability in high-dimensional, noisy, and heavily confounded observational data. More broadly, black-box time-series models (including neural ODE variants and deep sequence models) can capture rich temporal structure but are rarely compatible with bedside execution and often require substantial computational infrastructure, making them poor substitutes for guideline-style scoring systems when the deployment target is a human-executable protocol (Chen et al., 2018). Our work is complementary: we borrow the idea of structured hypothesis classes, but we focus on a constrained, checklist-oriented rule language that is explicitly aligned with clinical workflows. `AgentScore` is related in spirit to recent work that uses LLMs to propose structured candidates within search-and-verify or search-and-optimize loops, including FunSearch (Romera-Paredes et al., 2023), OPRO (Yang et al., 2024), LLM-SR (Shojaee et al., 2025), and LLM-based evolutionary optimization (Liu et al., 2025). At the same time, our setting differs in a crucial way: the hypothesis class is not an unconstrained search space, but one explicitly defined by clinical deployment requirements rather than purely predictive optimization.

**Large Language Models in Clinical Applications.** A rapidly growing literature studies LLMs for clinical documentation (note/discharge summary drafting), information extraction, triage and diagnostic suggestion, and medical question answering (Singhal et al., 2023; Pal et al., 2022). While these results indicate substantial medical knowledge and reasoning capacity, multiple studies emphasize that benchmark accuracy alone is insufficient for autonomous clinical decision-making: LLMs can hallucinate, exhibit instruction and context-order sensitivity, and remain difficult to calibrate and integrate into real workflows, with additional constraints from privacy and governance when patient data must be transmitted to external services (Hager et al., 2024; Williams et al., 2024). `AgentScore` differs fundamentally from direct LLM-based prediction:

the LLM is used only during development as a constrained proposal mechanism to explore a semantic rule space, while all acceptance is determined by deterministic, data-grounded verification (and a conservative eliminative plausibility filter). The deployed artifact is a static, deterministic unit-weighted checklist that requires no LLM at inference time, avoiding the latency, cost, and reliability risks of bedside LLM invocation and ensuring that final decisions are induced by validated rules on data rather than free-form model generation.

### C.3. Clinical model deployment and workflow integration

**Deployment challenges in clinical ML.**  A recurring theme in clinical machine learning is the gap between retrospective performance and real-world impact. Predictors trained on EHR data can fail under dataset shift, evolving clinical practice, missingness, and changes in coding or documentation. Practical deployment further requires robust integration into clinical workflows, governance, monitoring, and often regulatory review, all of which impose constraints beyond standard ML benchmarks. These challenges are widely documented and have motivated calls for rigorous evaluation, transparent reporting, and human-centered design in clinical decision support (Goldstein et al., 2016; Kelly et al., 2019; Hofer et al., 2020).

**Models aligned with guideline workflows.**  A promising direction is to align model outputs with existing clinical decision points rather than attempting to replace workflows end-to-end. In several domains, ML systems act as perception or measurement enhancers (e.g., image-based classification in dermatology, computational pathology, or radiology) (Kather et al., 2020; Esteva et al., 2017; Cao et al., 2023), while the downstream decision logic remains anchored in established clinical pathways and guidelines. This separation between *measurement* and *decision* can improve adoption: the model augments an upstream signal, and clinicians retain control over guideline-aligned thresholds and actions. This paradigm suggests that the most deployable ML systems may be those that explicitly target the model classes and interfaces used in routine practice.

**Interpretability versus deployability.**  Finally, it is important to distinguish interpretability from deployability. Interpretability concerns whether humans can understand the rationale for a model's outputs; deployability concerns whether the model can be executed reliably within the operational constraints of the target setting. A model can be interpretable yet non-deployable if it requires non-routine inputs, complex computations, or specialized infrastructure; conversely, a deployable guideline score may be less expressive yet succeed because it is auditable, memorable, and tightly coupled to actionable thresholds. This distinction motivates constraining the hypothesis class to *guideline-compatible* forms rather than optimizing flexible predictors and attempting to explain or simplify them post hoc (Rudin, 2019). `AgentScore` operationalizes this principle by searching over a clinically motivated rule language while enforcing hard structural constraints that ensure the learned artifact is executable at the bedside. This distinction is increasingly reflected in clinical AI reporting guidance, which separates transparency of model internals from suitability for real-world execution (Moons et al., 2025).

In summary, prior approaches are typically either deployable but manually designed (e.g., CURB-65), data-driven but not checklist-deployable (e.g., regression-to-points, RiskSLIM/FasterRisk, trees), or checklist-deployable but restricted to fixed candidate features (e.g., Optimal Checklists). In contrast, `AgentScore` is deployable-by-design and automatically constructs the clinically meaningful rules themselves (see Table 7).

*Table 7.* **At-a-glance comparison.** Interpretable vs. deployable vs. derived-rule construction capability, with representative examples.

| Approach | Interpretable | Checklist-deployable | Constructs derived rules | Example |
|---|---|---|---|---|
| Expert design | ✓ | ✓ | ✓ (manual) | CURB-65 |
| Regression + rounding | ✓ | partial | × | Framingham |
| RiskSLIM / FasterRisk | ✓ | partial | × | — |
| Optimal Checklists | ✓ | ✓ | × | $N$-of-$M$ checklist |
| Decision trees | ✓ | × | × | CART |
| AgentScore | ✓ | ✓ | ✓ (automated) | This work |

# D. Experimental details

## D.1. Datasets

We evaluate `AgentScore` on ten prediction tasks spanning five data sources:

1. **MIMIC-IV** (v3.1) (Johnson et al., 2023): A de-identified electronic health record (EHR) database from Beth Israel Deaconess Medical Center containing over 400,000 hospital admissions.

2. **eICU Collaborative Research Database** (v2.0) (Pollard et al., 2018): A multicenter critical care database comprising ICU stays from 208 hospitals across the United States.

3. **UK Cystic Fibrosis (CF) Registry**: A national registry containing annual longitudinal follow-up records for individuals with cystic fibrosis in the United Kingdom.

4. **Canadian Cystic Fibrosis Registry**: A national population-based registry used for external validation of CF mortality prediction.

5. **PhysioNet Challenge 2012** (Silva et al., 2012): A publicly available ICU mortality benchmark comprising 8,000 patient episodes from two hospitals.

**Task definitions.** Table 8 summarizes outcome definitions, prediction horizons, index times, and inclusion criteria. We provide additional clarifications below to ensure precise reproducibility.

**Observation windows.** For all time-series tasks, only measurements recorded strictly *before* the prediction horizon are used to prevent information leakage:

- MIMIC AF, HF, AKI, COPD, pneumonia (lung), lung cancer (cancer), and ICU mortality tasks use laboratory and vital-sign measurements up to **245 hours** from hospital admission or ICU intime.

- MIMIC length-of-stay tasks use measurements from the **first 24 hours** only.

- eICU tasks use measurements from the **first 6 hours** of ICU admission.

**Outcome construction.** Mortality outcomes correspond to binary in-hospital death unless otherwise specified. Length-of-stay (LOS) outcomes are defined using clinically meaningful thresholds: ICU LOS >3 days; LOS >5 days for COPD and lung cancer; and LOS >7 days for pneumonia. For cystic fibrosis cohorts, the outcome is defined as death within a fixed prediction horizon following the index annual visit.

**Missing data handling.** Time-series variables are imputed using forward fill; if no prior measurement exists within the observation window, backward fill is applied when later measurements are available. Median imputation is applied when no observed value is available. For baselines that do not operate on temporal data, the final observed value per variable is used.

**Train/validation/test splits.** We use 5-fold stratified cross-validation over trajectories (hospital admissions or ICU stays). Stratification preserves outcome prevalence across folds, and all splits are deterministic with a fixed random seed. For the CF cohort comparison, models are trained on the UK CF Registry and externally validated on the Canadian CF Registry to assess cross-population generalization. For ICU mortality external validation, models are trained on hospital A and evaluated on hospital B. Because fold-level splits are unavailable in this setting, results are aggregated over five independent random training seeds. For `AgentScore`, the training portion of each fold is further split into 80% used for rule construction and 20% used for internal validation during score selection. The baselines utilize the training data according to their standard respective optimization procedures.

**Statistical analysis.** We evaluate predictive performance using AUROC. For `AgentScore`, predicted probabilities are computed as

$$\hat{p} = \frac{\text{count}}{n_{\text{rules}}},$$

*Table 8.* **Task definitions.** Outcome definitions, prediction horizons, index times, and key inclusion criteria for each prediction task.

| Task | Outcome | Horizon | Index Time | Inclusion Criteria |
|---|---|---|---|---|
| MIMIC-AF | In-hospital mortality | Stay | Admission | Adults with AF (ICD-9: 42731; ICD-10: I48.*) |
| MIMIC-HF | In-hospital mortality | Stay | Admission | Adults with heart failure (ICD-9: 428.*; ICD-10: I50.*) |
| MIMIC-AKI | AKI development | Stay | Admission | Adults; AKI per ICD-9: 584.* or ICD-10: N17.* |
| MIMIC-COPD | LOS >5 days | Stay | Admission | Adults with COPD (ICD-9: 491.*, 492.*, 496; ICD-10: J43.*, J44.*) |
| MIMIC-Lung | LOS >7 days | Stay | Admission | Adults with pneumonia (ICD-9: 480–486; ICD-10: J12–J18) |
| MIMIC-Cancer | LOS >5 days | Stay | Admission | Adults with lung cancer (ICD-9: 162; ICD-10: C34.*) |
| eICU-Vaso | Vasopressor initiation | Delayed | ICU admission | Adults; excludes stays <1h |
| eICU-LOS | ICU LOS >3 days | Stay | ICU admission | Adults; excludes stays <1h |
| CF-UK | Mortality | 3 years | Annual visit | Patients with annual follow-up records |
| CF-CA | Mortality | Horizon-based | Annual visit | Patients with $\geq 2$ annual records |
| ICU-Mort. | In-hospital mortality | Stay | ICU admission | PhysioNet 2012 cohort |

where *count* denotes the number of satisfied rules. Statistical significance is assessed using paired tests across cross-validation folds, including both a paired t-test and a Wilcoxon signed-rank test. Two-sided alternatives are used with hypothesis $H_1$: `AgentScore` > baseline. In some folds, heavily regularized or discretized baselines fail to produce a non-degenerate predictor (e.g., all coefficients collapse to zero). In these cases, AUROC is conservatively set to 0.5 to enable paired comparisons without discarding folds.

*Table 9.* **Dataset summary statistics.** Patients denote unique individuals; observations denote total time-indexed records.

| Statistic | MIMIC AF | eICU Vaso | eICU LOS | MIMIC HF | MIMIC Lung | MIMIC Cancer | MIMIC COPD | MIMIC AKI | CF-CA | CF-UK | ICU-Mort. |
|---|---|---|---|---|---|---|---|---|---|---|---|
| Patients | 79,779 | 195,339 | 195,339 | 80,611 | 29,180 | 8,245 | 44,246 | 546,028 | 7,582 | 11,741 | 8,000 |
| Features | 74 | 60 | 60 | 74 | 74 | 74 | 74 | 75 | 107 | 132 | 42 |
| Observations | 8.5M | 195K | 195K | 9.2M | 451K | 114K | 614K | 38.3M | 144K | 58.7K | 381K |
| Pos. (%) | 5.4 | 10.7 | 25.3 | 5.0 | 46.2 | 44.6 | 42.0 | 13.4 | 32.0 | 4.9 | 14.0 |

**Feature binarization.** Score-learning methods (RiskSLIM, FasterRisk, and PLR variants) require binary indicator features to learn integer-valued scoring systems (Liu et al., 2022). For each continuous variable $x_{\cdot, j}$, we construct a set of binary threshold indicators $\tilde{x}_{\cdot, j, \theta} = \mathbb{I}[x_{\cdot, j} \leq \theta]$. When the number of unique values in column $j$ is small, we use all unique values as thresholds (excluding the maximum to avoid constant predictors). When the number of unique values exceeds a configurable cap (default: 1000), we instead use quantile-based thresholds at probabilities $\{1/q, 2/q, \ldots, (q-1)/q\}$ for $q = 1000$ to maintain computational tractability. Constant dummy columns (all zeros or all ones on the training set) are dropped. This *threshold-dummy transform* is fitted on training data and applied identically to test data to prevent leakage. Optimal Checklists automatically applies its own binarization/binning procedure.

### D.2. Baselines

We note that most existing score-learning or interpretable modeling approaches do not natively support the construction of unit-weighted $N$-of-$M$ clinical checklists as produced by `AgentScore`. Methods such as RiskSLIM, FasterRisk, and pooled piecewise-linear rule (PLR) variants optimize over sparse linear models with integer or real-valued coefficients, but do not enforce unit weights or checklist-style aggregation. AutoScore can generate integer-valued point systems, but the resulting scores are not bounded and may assign large cumulative point totals, complicating memorability and bedside use. Optimal Checklists is the closest exception: it learns unit-weighted $N$-of-$M$ checklists, but only over a fixed, pre-specified candidate feature set. It therefore does not address the complementary problem central to `AgentScore`: constructing clinically meaningful derived rules, such as ratios, temporal changes, thresholded trends, or logical compositions, before validating and assembling them into a deployable checklist.

To ensure a fair and meaningful comparison, we restrict all integer-based baselines to the coefficient set $\{-1, 0, +1\}$ and limit model sparsity to match the maximum number of rules used by `AgentScore`. This aligns the hypothesis classes as closely as possible while preserving each method's native optimization procedure.

**Small integer-based score-learning baselines.** We compare against established methods for learning sparse, interpretable scoring systems:

- **RiskSLIM** (Ustun & Rudin, 2015): A mixed-integer programming (MIP) approach for learning sparse linear models with integer coefficients. We use the CPLEX solver with a maximum runtime of 3000 seconds and an $L_0$ penalty $c_0 = 10^{-6}$. Features are standardized to zero mean and unit variance, and the threshold-dummy transform is optionally applied. Coefficients are constrained to $\{-1, 0, +1\}$, and the sparsity budget is matched to `AgentScore`.

- **FasterRisk** (Liu et al., 2022): A fast coordinate-descent–based algorithm for learning sparse risk scores. We use the official Python implementation with sparsity constraint $k = 6$ and coefficient bounds $\{-1, 0, +1\}$. Features are standardized prior to fitting, and the threshold-dummy transform is optionally applied via configuration.

- **Optimal Checklists** (Zhang et al., 2021): An integer-programming approach for learning sparse binary checklists. We use the official implementation with a sparsity constraint $N = 6$ (maximum number of selected checklist items) and optimize the decision threshold $M$ during training.

**AutoScore.** AutoScore (Xie et al., 2020) is a framework for automatically constructing point-based clinical scores. We use the reference R implementation with random forest–based feature selection (100 trees), equal-frequency discretization into four bins for continuous variables, and a maximum score of 100 points. AutoScore performs its own internal binning and does not rely on the shared threshold-dummy transform. While AutoScore produces integer-valued scores, the resulting point totals are not bounded to unit weights, leading to less compact and less checklist-like models.

**Linear and tree-based baselines.** We include standard interpretable machine-learning comparators operating on continuous features without binarization. Logistic regression is trained with feature standardization (zero mean, unit variance), median imputation for missing values, and the SAGA optimizer with a maximum of 2000 iterations. As a non-linear but still interpretable comparator, we include a CART decision tree with maximum depth four.

**PLR baselines.** Following Liu et al. (2022), we include pooled piecewise-linear rule (PLR) baselines, which expand continuous variables into collections of binary threshold indicators prior to fitting sparse linear models. This expansion uses the same threshold-dummy transform applied to other baselines.

Each PLR model is trained by solving an ElasticNet-regularized logistic regression problem over the expanded feature space, with mixing parameter $\alpha \in \{0, 0.1, \ldots, 1.0\}$ and regularization strength $C \in [10^{-4}, 10^2]$ on a logarithmic scale, yielding real-valued coefficients.

To obtain integer-valued scoring systems, we apply the rounding procedures introduced by Ustun & Rudin (2019), which convert real-valued PLR solutions into discrete coefficients in $\{-1, 0, +1\}$. For each dataset, we generate a pool of approximately 1,100 candidate PLR models (11 $\alpha$ values $\times$ 8 regularization settings $\times$ multiple random seeds) and apply multiple rounding strategies. The final model is selected as the rounded solution with the lowest logistic loss.

Specifically, we consider:

- *PLR-RD*: Direct rounding after truncating coefficients to $[-1, +1]$.

- *PLR-RDU*: Unit-weighted rounding (Burgess method), assigning coefficients based solely on sign.

- *PLR-RSRD*: Rescaling coefficients to unit magnitude prior to rounding.

- *PLR-Rand*: Randomized rounding based on fractional coefficient values.

- *PLR-RDP*: Sequential, loss-aware rounding that locally minimizes logistic loss.

- *PLR-RDSP*: Sequential rounding followed by discrete coordinate descent (25 iterations) to further reduce loss.

## D.3. `AgentScore` Hyperparameters

Table 10 summarizes the hyperparameters used for `AgentScore` across all experiments.

We set `auc_threshold`= 0.60 empirically as the lowest univariate AUROC that still justifies including a rule under a sparse rule budget. For diversification, `jaccard_threshold`= 0.9 filters near-duplicate rules (high overlap in covered positives); we allow an exception when a candidate improves AUROC by at least 0.01 over the incumbent. For scoring, we use a deterministic Youden-$J$ thresholding objective by default, but other objectives (e.g., prioritizing sensitivity or PPV) can be substituted depending on the desired operating point.

*Table 10.* **`AgentScore` hyperparameters.** Default values used across all experiments unless otherwise specified.

| Parameter | Value | Description |
|---|---|---|
| *Feature Agent* | | |
| `max_rules` | 6 | Maximum number of rules in final score |
| `iterations` | 100 | Number of LLM proposal iterations |
| `auc_threshold` | 0.60 | Minimum univariate AUROC to retain a candidate rule |
| `temperature` | 1.0 | LLM sampling temperature for diversity |
| `logic_depth` | 1 | Maximum logical composition depth (atomic single-condition rules) |
| *Score Diversification* | | |
| `jaccard_threshold` | 0.95 | Maximum Jaccard similarity on positive cases |
| `min_pos_gain` | 0.001 | Minimum AUROC gain to accept a similar rule |
| *Scoring Agent* | | |
| `refine_steps` | 10 | Number of refinement iterations |
| `objective` | Youden | Threshold selection objective (Youden's $J$) |

## D.4. Compute resources

Experiments were conducted on a shared Linux workstation with dual AMD EPYC 7713 CPUs (128 physical cores), 1 TB of system memory, and a single NVIDIA RTX 6000 Ada GPU (48 GB VRAM), using CUDA 11.5. We use GPT-5 (version 2025-08-07), GPT-4o (version 2024-11-20) and DeepSeek V3.2 (version 2026-05-01-preview) via API calls as provided on Microsoft Azure.

# E. Extended results

## E.1. Formal Optimization Landscape and Complexity

**Problem Formulation.** Let $\mathcal{D} = \{(x_i, y_i)\}_{i=1}^{N}$ be a dataset where $x_i \in \mathbb{R}^d$ and $y_i \in \{0, 1\}$. We define a rule grammar $\mathcal{G}$ generating a universe of logical predicates $\mathcal{R}_{\text{univ}}$. The objective is to select a subset $S \subset \mathcal{R}_{\text{univ}}$ to form a unit-weighted score $f_S(x) = \sum_{r \in S} r(x)$ that maximizes a utility metric $\mathcal{J}$.

$$\max_{w \in \{0,1\}^{|\mathcal{R}_{\text{univ}}|}, k \in \mathbb{Z}} \mathcal{J}(w, k; \mathcal{D}) \quad \text{s.t.} \quad \|w\|_0 \leq M_{\max}, \quad \hat{y} = \mathbb{I}\left(\sum_j w_j r_j(x) \geq k\right). \tag{2}$$

Here $\mathcal{J}$ is a utility metric (e.g., AUROC) and $\|w\|_0$ is the $\ell_0$ pseudo-norm.

**Why Standard Solvers Fail.**

1. **The "Generation–Selection" Gap:** State-of-the-art interpretable solvers such as RiskSLIM and FasterRisk act as *selectors*, requiring a pre-computed feature matrix $X \in \{0, 1\}^{N \times |\mathcal{R}_{\text{univ}}|}$. They cannot generate semantic rules dynamically; any clinically meaningful derived constructs (ratios, trends, shallow logic) must be manually engineered and materialized as columns *a priori*.

2. **Failure of Continuous Relaxation (e.g., Lasso):** Relaxing $w \in \{0, 1\}^{|\mathcal{R}_{\text{univ}}|}$ to continuous weights $w \in \mathbb{R}^{|\mathcal{R}_{\text{univ}}|}$ introduces a substantial **integrality gap**: continuous relaxations induce fractional solutions, and rounding them can change the induced decision boundary and utility relative to the discrete optimum. In checklist learning, where both the score and the operating threshold are discrete, such rounding effects are often amplified.

3. **Failure of Classical Heuristics (Genetic Algorithms, SA):** Standard heuristic searches struggle with the semantic structure of the rule space:

- *Undefined Metric Space:* Crossover operators in Genetic Algorithms require a meaningful metric space. It is unclear how to interpolate between "Age $> 65$" and "Lactate $< 2.0$".
- *Sparse Fitness Landscape:* A random mutation to a complex rule (e.g., changing a temporal window from 24h to 1h) often yields a rule with AUROC $\approx 0.5$. Without semantic guidance, random search (and simulated annealing) wastes the large majority of evaluations on statistically irrelevant candidates.

**Combinatorial and Physical Intractability (order-of-magnitude).** The necessity of `AgentScore` is driven by the sheer scale of the grammar-induced search space. Below we provide an *order-of-magnitude* breakdown of the candidate space size for a representative clinical task ($p = 50$ variables, $N = 5 \times 10^5$ patients). For clarity, we omit several additional rule families (e.g., some higher-arity compositions and additional temporal operators); including them would only increase the space further.

*1. Primitive Rules:* With $T = 20$ quantile thresholds and range constraints, a crude count gives

$$|\mathcal{R}_{\text{prim}}| \approx p \times (2T + T^2) \approx 50 \times 440 \approx 2.2 \times 10^4.$$

*2. Compositional Rules:* Allowing depth-1 logical operators (AND/OR) between pairs of primitives yields, up to constants,

$$|\mathcal{R}_{\text{comp}}| \approx 2 \cdot \binom{|\mathcal{R}_{\text{prim}}|}{2} \approx \mathcal{O}\big(|\mathcal{R}_{\text{prim}}|^2\big) \approx (2.2 \times 10^4)^2 \approx 4.8 \times 10^8.$$

*3. Additional Variants (Temporal + Ratios):* Introducing simple temporal summaries (e.g., $W = 4$ windows $\times$ 3 stats = 12 variants) and a restricted set of arithmetic ratios/differences over variable pairs ($\binom{50}{2} \approx 1225$) increases the candidate universe by large multiplicative factors. An order-of-magnitude approximation is

$$|\mathcal{R}_{\text{univ}}| \approx |\mathcal{R}_{\text{comp}}| \times (1 + 12_{\text{temporal}}) \times (1 + 1225_{\text{ratios}}) \approx (4.8 \times 10^8) \times 13 \times 1226 \approx \mathbf{7.6 \times 10^{12}}.$$

**The Physical Barriers.** *The Memory Wall:* Instantiating a full binary feature matrix $X \in \{0,1\}^{N \times |\mathcal{R}_{\text{univ}}|}$ for matrix-based solvers would require storing on the order of

$$N \times |\mathcal{R}_{\text{univ}}| \approx (5 \times 10^5) \times (7.6 \times 10^{12}) \approx 3.8 \times 10^{18}$$

binary entries. Even under ideal bit-packing (1 bit per entry), this is

$$3.8 \times 10^{18} \text{ bits} \approx 4.75 \times 10^{17} \text{ bytes} \approx \mathbf{475} \text{ PB}.$$

In practice, sparse/dense representations and metadata overhead would increase this further. This creates a hard physical constraint: the full rule matrix cannot be pre-computed; features must be generated on-the-fly.

*The Time Wall:* Even with on-the-fly generation, assuming a realistic evaluation time of $\tau = 0.1$s per rule (including temporal feature extraction over $N = 5 \times 10^5$ patients),

$$\text{Time} \approx |\mathcal{R}_{\text{univ}}| \times \tau \approx 7.6 \times 10^{12} \times 0.1\text{s} = 7.6 \times 10^{11}\text{s} \approx \mathbf{24{,}000} \text{ years}.$$

**Conclusion.** The problem space is too large for enumeration (Time Wall), too large for standard matrix-based solvers (Memory Wall), and ill-suited for gradient-based or random heuristics (integrality and semantic gaps). This motivates `AgentScore`: using an LLM to learn a proposal distribution $P_\theta(r)$ that navigates the semantic structure of the rule space, alleviating the "cold start" problem of finding high-utility rules in a sparse combinatorial landscape.

### E.2. Extended experimental results and ablations

Table 11 provides extended results for `AgentScore` and the pooled PLR variants omitted from the main text for readability. These PLR baselines evaluate post-hoc coefficient modification strategies for logistic models and consistently underperform

`AgentScore`, illustrating that compact deployable scores are not obtained simply by fitting a flexible model and modifying its coefficients after training.

To disentangle rule construction from downstream score assembly, we additionally evaluate representation-controlled baselines that are given access to the same validated rule pool generated by `AgentScore`. These ablations isolate whether performance gains arise solely from access to clinically structured derived rules or also from the proposal–validation–assembly procedure used to construct the final checklist. FasterRisk applied to the `AgentScore` rule pool approaches the full pipeline but remains lower on average, while the MIP-based Optimal Checklist solver performs competitively on some tasks but is less stable across datasets. Greedy selection performs substantially worse, indicating that naive local assembly is insufficient even when the candidate rules are clinically structured. Overall, these results suggest that the gains of `AgentScore` are not attributable only to feature construction: effective validation and checklist assembly are also important for obtaining deployable unit-weighted scores.

*Table 11.* **Extended model performances and representation-controlled ablations.** Entries are mean ± std AUROC. Rows marked `AgentScore` + solver evaluate classical score-learning or selection procedures on the same validated rule pool generated by `AgentScore`, isolating the effect of rule construction from downstream search and assembly. For each dataset, the best-performing score-based or checklist method is shown in **bold**.

| Method | MIMIC AF | MIMIC COPD | MIMIC HF | MIMIC AKI | MIMIC Cancer | MIMIC Lung | eICU LOS | eICU Vaso | Mean |
|---|---|---|---|---|---|---|---|---|---|
| Pooled PLR (Rand) | 0.70 ± 0.01 | 0.58 ± 0.00 | 0.70 ± 0.01 | 0.76 ± 0.00 | 0.55 ± 0.01 | 0.50 ± 0.00 | 0.62 ± 0.00 | 0.50 ± 0.00 | 0.62 ± 0.09 |
| Pooled PLR (RDP) | 0.70 ± 0.01 | 0.58 ± 0.00 | 0.70 ± 0.01 | 0.76 ± 0.00 | 0.55 ± 0.01 | 0.50 ± 0.00 | 0.62 ± 0.00 | 0.50 ± 0.00 | 0.62 ± 0.09 |
| Pooled PLR (RDSP) | 0.70 ± 0.01 | 0.58 ± 0.00 | 0.70 ± 0.01 | 0.76 ± 0.00 | 0.55 ± 0.01 | 0.50 ± 0.00 | 0.62 ± 0.00 | 0.50 ± 0.00 | 0.62 ± 0.09 |
| Pooled PLR (RD) | 0.70 ± 0.01 | 0.58 ± 0.00 | 0.70 ± 0.01 | 0.76 ± 0.00 | 0.55 ± 0.01 | 0.50 ± 0.00 | 0.50 ± 0.00 | 0.50 ± 0.00 | 0.60 ± 0.10 |
| Optimal Checklists (raw + rules) | 0.72 ± 0.07 | 0.60 ± 0.05 | 0.68 ± 0.10 | 0.78 ± 0.04 | **0.59 ± 0.01** | 0.62 ± 0.01 | 0.66 ± 0.00 | 0.50 ± 0.00 | 0.64 ± 0.04 |
| `AgentScore` + FasterRisk | 0.79 ± 0.02 | 0.63 ± 0.00 | 0.77 ± 0.03 | 0.77 ± 0.02 | 0.58 ± 0.01 | 0.58 ± 0.01 | 0.65 ± 0.01 | 0.74 ± 0.01 | 0.69 ± 0.08 |
| `AgentScore` + MIP (Optimal Checklist solver) | 0.76 ± 0.03 | **0.64 ± 0.02** | 0.69 ± 0.08 | 0.75 ± 0.04 | **0.59 ± 0.01** | 0.59 ± 0.01 | 0.66 ± 0.00 | 0.50 ± 0.08 | 0.65 ± 0.09 |
| `AgentScore` + Greedy Selection | 0.56 ± 0.07 | 0.52 ± 0.05 | 0.55 ± 0.06 | 0.61 ± 0.06 | 0.50 ± 0.00 | 0.50 ± 0.00 | 0.50 ± 0.00 | 0.50 ± 0.08 | 0.53 ± 0.05 |
| `AgentScore` (full pipeline) | **0.81 ± 0.01** | 0.63 ± 0.03 | **0.79 ± 0.01** | **0.79 ± 0.02** | **0.59 ± 0.01** | **0.63 ± 0.00** | **0.67 ± 0.00** | **0.76 ± 0.00** | **0.71 ± 0.08** |

## E.3. Effect of different LLM backbones

`AgentScore` relies on a language model to navigate the rule space by proposing semantically plausible candidates. While the downstream evaluation, acceptance, and selection procedures are fully deterministic, the quality of the proposal distribution depends on the underlying LLM. We therefore assess the sensitivity of `AgentScore` to the choice of language model backbone. For the main experiments, we use GPT-5, a frontier closed-source model. To evaluate robustness to model choice, cost, and openness, we additionally consider a strong open-source alternative (DeepSeek V3.2) as well as a computationally cheaper proprietary model (GPT-4o).

Across all datasets, GPT-5 yields the strongest overall performance; however, the performance gap to GPT-4o and DeepSeek V3.2 is small, with all models achieving competitive average AUROC (Figure 4). These results highlight two practical implications. First, the framework remains effective under substantially lower-cost or open-source model choices, improving accessibility and reproducibility. Second, because the LLM is used solely as a proposal mechanism and never bypasses deterministic statistical validation, the proposal quality may improve as language models advance, while deterministic validation continues to constrain accepted rules. Future gains in LLM reasoning quality are therefore likely to translate directly into more efficient search and higher-quality guideline-style scoring systems, without requiring changes to the underlying optimization framework.

## E.4. Risk calibration and score monotonicity

For clinical deployment, performance metrics such as AUROC are necessary but not sufficient. In practice, clinicians do not treat scoring systems as purely binary classifiers. Instead, scores are used to *contextualize risk*: borderline scores may prompt individualized clinical judgment, while very high scores often trigger heightened vigilance, escalation, or additional review even when the clinician's prior concern is low. Consequently, a clinically usable scoring system must exhibit *risk monotonicity*, such that higher scores correspond to systematically higher event rates.

To assess this property, we analyze outcome prevalence as a function of the discrete score value produced by `AgentScore`. Across all datasets, we observe a consistent and approximately monotonic increase in empirical risk with increasing score (Figure 5). This indicates that the learned unit-weighted checklists preserve ordinal risk structure and support meaningful risk stratification beyond a single operating threshold.

The only apparent deviation from strict monotonicity occurs in the MIMIC Lung dataset, where the score bin of zero exhibits a slightly higher positive rate than the score bin of one. This effect is attributable to extreme data sparsity: only 13 patients

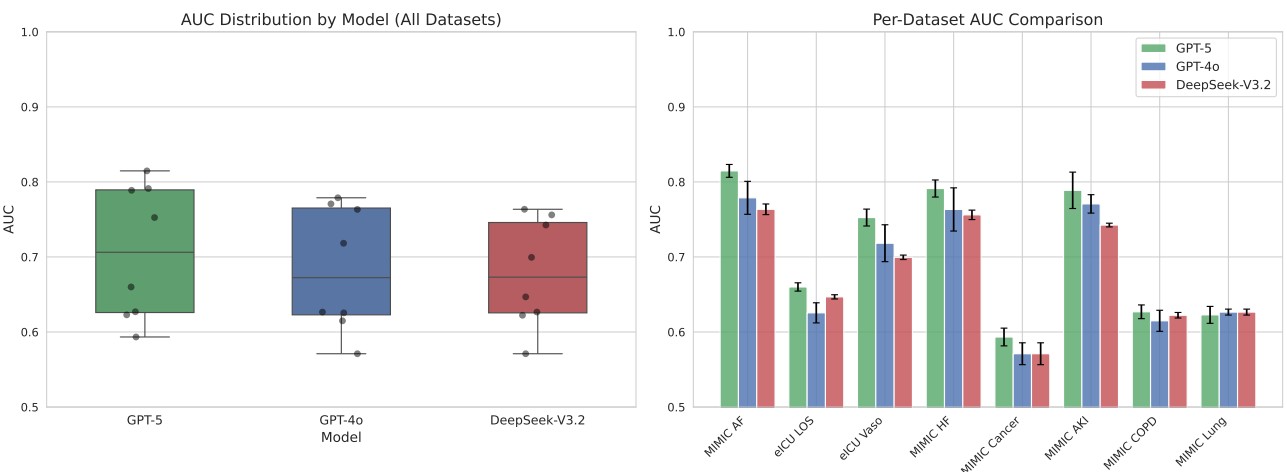

*Figure 4.* **Effect of LLM backbone choice.** Performance of `AgentScore` across different language-model backbones. While GPT-5 attains the strongest results on average, GPT-4o and DeepSeek V3.2 exhibit comparable performance trends, suggesting limited sensitivity to the specific LLM used for rule proposal.

fall into the zero-score bin, compared to several thousand patients in all other bins. When accounting for sampling variability, the overall risk trend remains monotone.

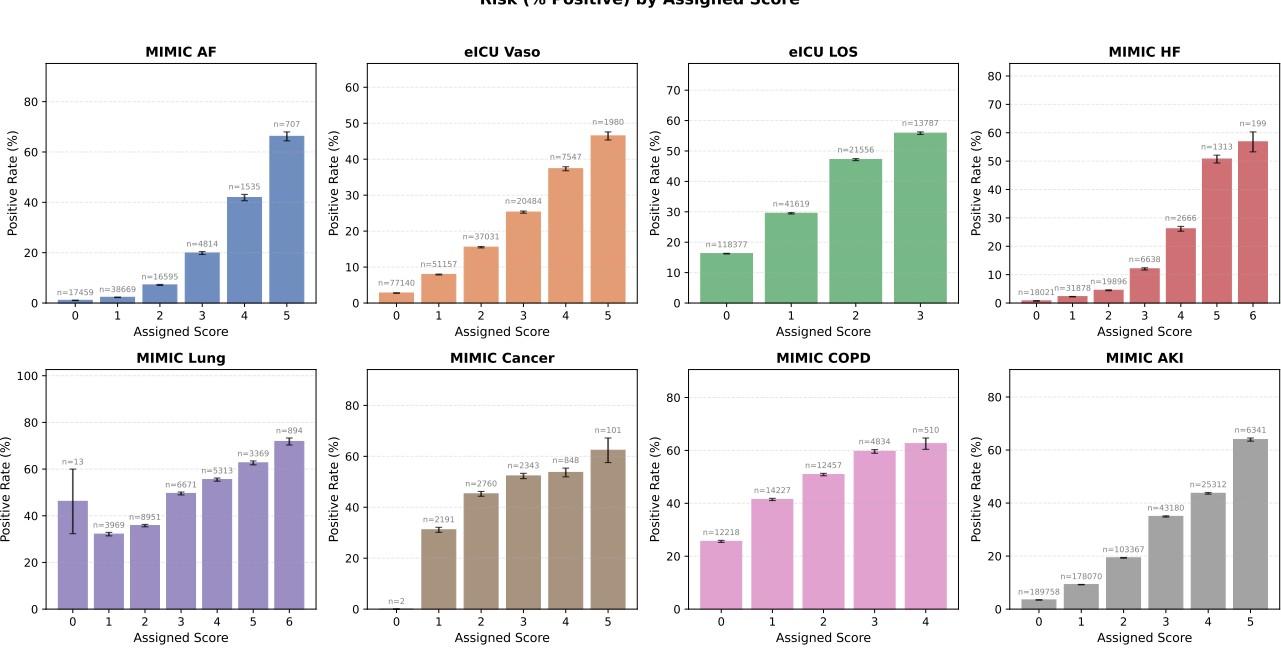

*Figure 5.* **Risk monotonicity across score values.** Empirical outcome prevalence (percentage positive) as a function of the discrete `AgentScore` value.

## E.5. Effect of guideline size

Throughout the paper, we restrict the maximum number of rules to $M_{\max} = 6$, reflecting a conservative upper bound for checklists that remain easily memorizable and executable at the bedside while achieving competitive predictive performance. In practice, however, clinical stakeholders may prefer even smaller scores, motivating an explicit trade-off between model complexity and accuracy.

To characterize this trade-off, we vary the maximum allowed number of rules and evaluate predictive performance as a function of checklist size. Across all datasets, performance increases smoothly and predictably with the number of included rules (Figure 6), yielding a clear accuracy–complexity Pareto frontier. Notably, in many tasks, strong performance is already achieved with only four or five rules, with diminishing returns beyond this point.

These results provide clinicians and guideline developers with direct control over deployability: additional rules can be retained when modest gains in accuracy are clinically meaningful, or omitted to maximize simplicity, memorability, and ease of use. This flexibility highlights the practical advantage of learning within a strictly constrained, unit-weighted checklist model class.

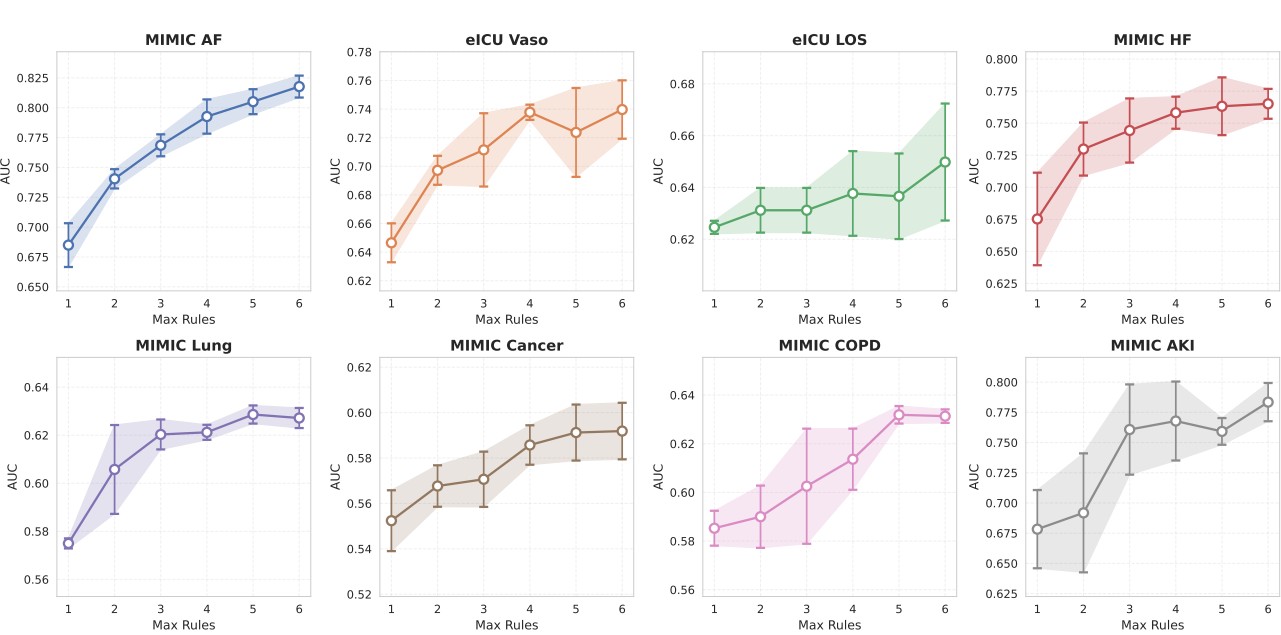

*Figure 6.* **Accuracy–complexity trade-off.** AUROC as a function of the maximum number of rules allowed in the checklist. Performance improves with rule set size, illustrating a clear deployability–accuracy Pareto frontier.

**Statistical analysis.** We evaluate predictive performance using AUROC. For `AgentScore`, predicted probabilities are computed as

$$\hat{p} = \frac{\text{count}}{n_{\text{rules}}},$$

where *count* denotes the number of satisfied rules. Statistical significance is assessed using paired tests on fold-level AUROC values across the 8 selected datasets (5 folds each; $n = 40$ paired observations per comparison), including a paired t-test and a Wilcoxon signed-rank test. Tests are two-sided, and $p$-values are corrected for multiple comparisons using Holm–Bonferroni. We additionally report bootstrap 95% confidence intervals for mean differences and Cohen's $d$ effect sizes. If a method is missing for a fold/dataset, AUROC is set to 0.5 to preserve pairing.

### E.6. Privacy Analysis

`AgentScore` reduces direct data exposure by restricting the LLM to feature metadata and aggregate evaluation statistics rather than raw patient-level records. However, this interface does not provide a formal privacy guarantee, especially when external API-based models are used: aggregate statistics, tool-interaction transcripts, or the final selected checklist may still leak information in principle. This risk can be further reduced by using locally hosted models and standard access controls on the tool interface.

To empirically assess leakage risk, we perform a membership-inference-style privacy audit. For each dataset, we run a no-LLM oracle that simulates `AgentScore`-style rule selection on paired held-out evaluation worlds differing by one target-vs-donor substitution. From each run, we construct attack features from either the interaction transcript, the final

*Table 12.* **Statistical comparison of AgentScore vs. score-based baselines.** Two-sided paired tests on fold-level AUROC values ($n =$ tasks $\times$ 5 folds). $p$-values corrected using Holm-Bonferroni.

| Baseline | $n$ | $\Delta$ AUROC | 95% CI | $p$ (t-test) | $p$ (Wilcoxon) | Cohen's $d$ |
|---|---|---|---|---|---|---|
| Optimal Checklists | 40 | **+0.086** | [+0.052, +0.120] | $< 0.001^{**}$ | $< 0.001^{**}$ | +0.78 |
| PLR (rd) | 40 | **+0.109** | [+0.087, +0.133] | $< 0.001^{**}$ | $< 0.001^{**}$ | +1.47 |
| PLR (rand) | 40 | **+0.094** | [+0.073, +0.118] | $< 0.001^{**}$ | $< 0.001^{**}$ | +1.26 |
| PLR (rdp) | 40 | **+0.094** | [+0.073, +0.118] | $< 0.001^{**}$ | $< 0.001^{**}$ | +1.26 |
| PLR (rdsp) | 40 | **+0.094** | [+0.073, +0.118] | $< 0.001^{**}$ | $< 0.001^{**}$ | +1.26 |
| PLR (rsrd) | 40 | **+0.066** | [+0.054, +0.079] | $< 0.001^{**}$ | $< 0.001^{**}$ | +1.67 |
| PLR (rdu) | 40 | **+0.058** | [+0.044, +0.072] | $< 0.001^{**}$ | $< 0.001^{**}$ | +1.26 |
| RiskSLIM | 40 | **+0.046** | [+0.033, +0.060] | $< 0.001^{**}$ | $< 0.001^{**}$ | +1.07 |
| FasterRisk | 40 | **+0.031** | [+0.019, +0.042] | $< 0.001^{**}$ | $< 0.001^{**}$ | +0.81 |

selected rules, or both. We then train a balanced logistic-regression attacker on generated attack samples using a grouped 80/20 split by target identity, and report attack AUROC, advantage $(2 \cdot \text{AUROC} - 1)$, and TPR at fixed low false-positive rates.

We consider three attacker settings. In the *Transcript only* setting, the attacker observes only the agent's tool- interaction history. In the *Final only* setting, the attacker observes only the final checklist. In the *Strong* setting, the attacker observes both the transcript and the final checklist. Across the eight MIMIC and eICU tasks, attack performance remains near chance in all settings. The strongest attacker achieves mean AUROC $0.509 \pm 0.044$, mean advantage $+0.018$, and TPR $0.055$ at $1\%$ FPR. The final-only and transcript-only settings yield similarly low leakage, with mean AUROCs of $0.507 \pm 0.056$ and $0.501 \pm 0.021$, respectively. These results indicate low observed leakage under our audit protocol. However, they should not be interpreted as a formal privacy guarantee (e.g., differential privacy), and they do not rule out stronger attacks outside the evaluated threat model.

*Table 13.* **Membership-inference-style privacy audit.** Attack performance is averaged across the eight MIMIC and eICU tasks.

| Attack setting | Mean AUROC | Mean advantage | TPR @ 1% FPR |
|---|---|---|---|
| Strong | $0.509 \pm 0.044$ | $+0.018$ | 0.055 |
| Final only | $0.507 \pm 0.056$ | $+0.015$ | 0.048 |
| Transcript only | $0.501 \pm 0.021$ | $+0.002$ | 0.016 |

### E.7. Robustness to missing measurements

To evaluate robustness under partial observability, we randomly drop input measurements at test time and compare `AgentScore` against Optimal Checklists on the two external-validation tasks. Across both tasks, performance decreases as missingness increases, with task-dependent behavior. On ICU mortality, Optimal Checklists performs slightly better under full data, while `AgentScore` remains competitive and both methods degrade under increasing missingness. On CF mortality, `AgentScore` substantially outperforms Optimal Checklists across all missingness levels. Overall, these results suggest that robustness to missing measurements is task-dependent, but `AgentScore` remains competitive under missingness and compares favorably across the evaluated external settings.

*Table 14.* **Robustness to randomly missing measurements at test time.** AUROC is reported as mean $\pm$ std across runs under full data, 10% random feature dropout, and 25% random feature dropout.

| Dataset | Method | Full | Drop 10% | Drop 25% |
|---|---|---|---|---|
| CF Mortality | `AgentScore` | **$0.601 \pm 0.014$** | **$0.591 \pm 0.012$** | **$0.576 \pm 0.010$** |
| CF Mortality | Optimal Checklists | $0.454 \pm 0.113$ | $0.461 \pm 0.091$ | $0.466 \pm 0.069$ |
| ICU Mortality | `AgentScore` | $0.738 \pm 0.031$ | $0.717 \pm 0.034$ | $0.688 \pm 0.034$ |
| ICU Mortality | Optimal Checklists | **$0.759 \pm 0.008$** | **$0.746 \pm 0.010$** | **$0.710 \pm 0.008$** |

### E.8. Scoring system analysis

We analyze the structure of the scoring systems produced by `AgentScore` across a full cross-validation run. Concretely, for each dataset we collect the final checklist generated in each of the five folds and summarize the distribution of rule families induced by the rule grammar (Appendix A). Figure 7 reports the resulting composition statistics, quantifying how often the learned checklists rely on different rule types, demonstrating that the framework consistently leverages multiple rule families rather than collapsing to a single template. To illustrate the learned artifacts, Table 15 presents four representative `AgentScore` checklists. Each checklist is unit-weighted and executable as an $N$-of-$M$ procedure, where the total score is the number of satisfied rules and the decision threshold $K$ is selected on validation data. These examples highlight the intended output form: compact, auditable rule sets that combine simple threshold predicates with occasional shallow conjunctions and clinically interpretable derived rules (e.g., ratios), while remaining compatible with bedside execution.

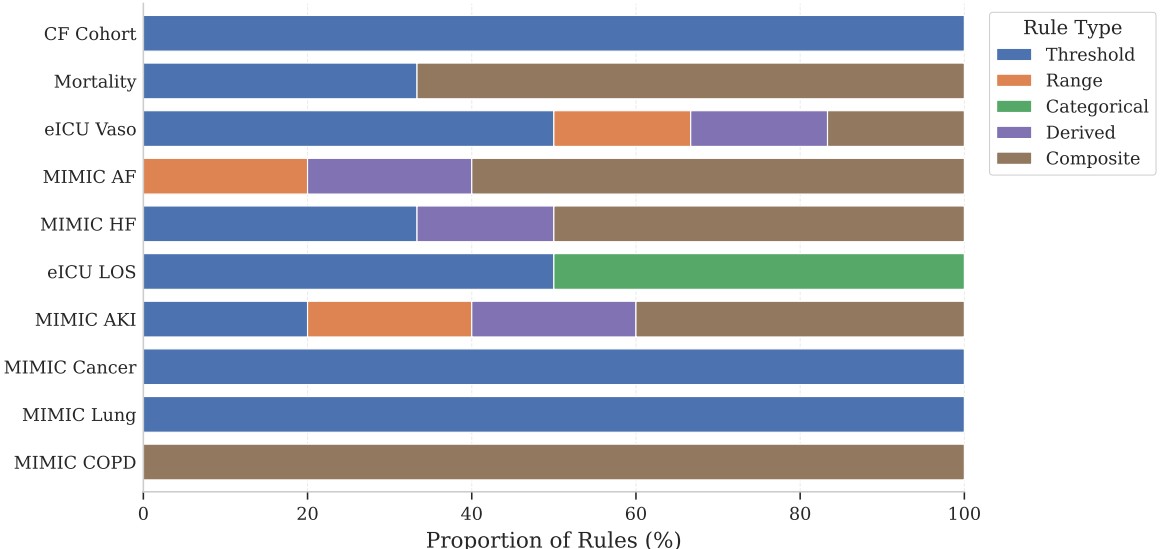

*Figure 7.* **Rule type diversity across cross-validation.** Distribution of rule families (as defined by the rule grammar) appearing in the final `AgentScore` checklists across five folds per dataset. Bars report the frequency with which each rule type is used, showing that learned scores draw on multiple rule families rather than collapsing to a single template.

### E.9. Clinician review of scoring system deployability

We conducted a structured expert review to assess the face validity, deployability, and usability of scoring systems generated by `AgentScore` compared to a strong baseline. This section provides full methodological details, statistical analyses, and response distributions to complement the summary in Section 5.3.

**Participants.**  A panel of $N = 18$ practicing clinicians participated in the study. Participants were recruited from six countries across three continents and represented diverse clinical specialties. The sample was predominantly experienced: 8 participants (44.4%) reported 10+ years of clinical experience, 8 (44.4%) reported 6–10 years, and 2 (11.1%) reported 0–2 years. Overall, 89% of participants had at least 6 years of clinical experience.

**Study design.**  We employed a blinded, randomized paired-comparison design. Participants evaluated four representative scoring systems generated by `AgentScore` (labeled "Model B") alongside matched outputs from `FasterRisk` (labeled "Model A"), a state-of-the-art integer score-learning baseline. To isolate structural and usability considerations from predictive performance, participants were explicitly instructed to assume equal discrimination and to focus exclusively on rule structure, feature choice, and bedside deployability.

**Survey instrument.**  The survey comprised 10 questions organized into three blocks:

**Block 1: Demographics and general attitudes (Q1–Q6).**

*Table 15.* **Representative `AgentScore` checklists (2×2 panel).** Each satisfied rule contributes +1 point. The total score is the number of satisfied rules; predict positive if $S(\mathbf{x}) \geq \tau$, where $\tau$ is selected on validation data. **Quantile thresholds.** Quantile-based cutpoints are common in clinical medicine (e.g., troponin above the 99th percentile upper reference limit; percentile-based definitions of "normal" ranges). For deployment, any quantile expression $Q_q(\cdot)$ is converted to a fixed numeric threshold estimated on a reference cohort (or precomputed and reported in guideline units); we retain the symbolic $Q_q$ notation here to show the original rule form produced by the system.

**eICU Prolonged ICU LOS (N-of-6)**
(Prolonged ICU Stay > 3 days Checklist)

| Checklist rule (satisfied?) | Points |
|---|---|
| GCS Verbal < 5 | +1 |
| GCS Eyes < 4 | +1 |
| GCS Motor < 6 | +1 |
| Hematocrit < 35% | +1 |
| Heart rate ≥ 85 bpm | +1 |
| Mean BP < 65 mmHg | +1 |
| **Total score** | **0–6** |
| **High-risk threshold** | $\mathbf{S(x) \geq 3}$ |

**MIMIC-IV HF Mortality (N-of-6)**
(HF In-Hospital Mortality N-of-M)

| Checklist rule (satisfied?) | Points |
|---|---|
| $HCO_3^-/Cl^- < 0.2$ | +1 |
| WBC ≥ 10 **and** $HCO_3^- \leq 25$ | +1 |
| BUN ≥ 30 | +1 |
| 1.5 ≤ Cr ≤ 5.0 | +1 |
| Admission is emergency **or** urgent | +1 |
| 10 ≤ WBC ≤ 20 | +1 |
| **Total score** | **0–6** |
| **High-risk threshold** | $\mathbf{S(x) \geq 3}$ |

**eICU Vasopressor (N-of-4)**
(Shock & Organ Dysfunction Checklist)

| Checklist rule (satisfied?) | Points |
|---|---|
| Intubated or mechanical ventilation | +1 |
| Heart rate / mean BP ≥ 1.5 | +1 |
| BUN ≥ 25 mg/dL | +1 |
| WBC ≥ $Q_{0.75}$(WBC) | +1 |
| **Total score** | **0–4** |
| **High-risk threshold** | $\mathbf{S(x) \geq 2}$ |

**MIMIC-IV AKI (N-of-4)**
(Renal–Anemia–Acuity Checklist)

| Checklist rule (satisfied?) | Points |
|---|---|
| Cr ≥ $Q_{0.75}$(Cr) | +1 |
| 0 ≤ Hematocrit ≤ 30 | +1 |
| ER admission **and** BUN ≥ 20 | +1 |
| BUN/Cr ≥ 20 | +1 |
| **Total score** | **0–4** |
| **High-risk threshold** | $\mathbf{S(x) \geq 2}$ |

- **Q1** (Demographics): Years of clinical experience (0–2, 3–5, 6–10, 10+).

- **Q2** (5-point Likert): "I would trust a clinical scoring system generated by AI."

- **Q3** (5-point Likert): "I would trust a clinical scoring system generated by AI *if it had been externally validated on 50,000 patients*."

- **Q4** (5-point Likert): "Memorability is important for clinical scoring systems used at the bedside."

- **Q5** (5-point Likert): "I would be willing to use a unit-weighted checklist (each item worth 1 point) in clinical practice."

- **Q6** (5-point Likert): "I would be willing to use a scoring system that requires mental arithmetic (e.g., multiplying values by 2, 3, or 5) at the bedside."

**Block 2: Paired comparisons across four clinical tasks (Q7–Q9).** For each of four clinical prediction tasks (ICU mortality, sepsis risk, respiratory failure, cardiac events), participants were shown two scoring systems (Model A and Model B) and asked:

- **Q7**: "Which scoring system better matches guideline-style clinical reasoning?"

- **Q8**: "Which scoring system would be easier to apply at the bedside under time pressure?"

- **Q9**: "Assuming both have been externally validated, which would you prefer to deploy?"

**Block 3: Overall preference (Q10).**

- **Q10**: "Overall, which scoring-system *style* would you prefer for clinical deployment?" (Model A / Model B / Neither).

**Statistical analysis**    All analyses were pre-specified. For Likert-scale items (Q2–Q6), we report means, 95% confidence intervals, and one-sample $t$-tests against the neutral midpoint (3). For the trust comparison (Q2 vs. Q3), we used a paired $t$-test and report Cohen's $d$ as an effect size. For pairwise preference questions (Q7–Q9), we aggregated responses across the 4 clinical tasks ($n = 72$ judgments per question) and tested whether the proportion preferring Model B exceeded 50% using one-sided binomial tests; we report Cohen's $h$ as an effect size for proportions. For overall preference (Q10), we used a chi-square goodness-of-fit test against a uniform distribution and a binomial test comparing Model B to Model A (excluding "Neither" responses). All $p$-values are reported without correction for multiple comparisons; conclusions are robust to Bonferroni correction.

**Trust in AI-generated scores.**    Baseline trust in AI-generated scores was neutral to low ($M = 2.72$, 95% CI $[2.13, 3.31]$, $t(17) = -0.92$, $p = 0.37$). However, trust increased substantially when large-scale external validation was specified ($M = 4.39$, 95% CI $[4.16, 4.62]$, $t(17) = 11.75$, $p < 10^{-8}$). The within-subject increase was statistically significant ($\Delta M = +1.67$, SD $= 1.37$; paired $t(17) = 5.15$, $p < 10^{-4}$) with a large effect size (Cohen's $d = 1.21$).

**Deployability preferences.**    Clinicians expressed strong willingness to use unit-weighted checklists ($M = 4.33$, 95% CI $[3.91, 4.75]$, $t(17) = 6.23$, $p < 10^{-5}$) and moderate willingness to use scores requiring mental arithmetic ($M = 3.89$, 95% CI $[3.58, 4.20]$, $t(17) = 5.58$, $p < 10^{-4}$). Importance of memorability was rated as neutral to high on average ($M = 3.22$, $p = 0.45$).

**Pairwise comparisons.**    Across 72 judgments per question (18 clinicians $\times$ 4 tasks), participants significantly preferred `AgentScore` (Model B) over `FasterRisk` (Model A) on all three dimensions:

- **Q7 (Guideline reasoning):** 85% preferred Model B (95% CI $[76\%, 92\%]$; binomial $p < 10^{-9}$, Cohen's $h = 0.77$).

- **Q8 (Bedside ease):** 71% preferred Model B (95% CI $[61\%, 80\%]$; binomial $p < 10^{-3}$, Cohen's $h = 0.43$).

- **Q9 (Deployment preference):** 81% preferred Model B (95% CI $[71\%, 89\%]$; binomial $p < 10^{-7}$, Cohen's $h = 0.66$).

**Within-respondent consistency.**    To assess reliability, we computed the proportion of respondents who gave the same answer across all four clinical tasks: 61% for Q7, 33% for Q8, and 50% for Q9. The moderate consistency for Q8 suggests that bedside-ease judgments are more task-dependent than guideline-alignment judgments.

**Overall preference.**    When asked for their overall preferred scoring-system style for clinical deployment, 12/18 clinicians (67%) selected Model B (`AgentScore`), 1/18 (6%) selected Model A (`FasterRisk`), and 5/18 (28%) reported no preference. The distribution differed significantly from uniform ($\chi^2(2) = 10.33$, $p = 0.006$). Excluding "Neither" responses, 12/13 (92%) preferred Model B over Model A (binomial $p = 0.002$).

**Qualitative observations.**    In free-text comments, clinicians noted that non-unit weights (e.g., $+3$, $+5$) in `FasterRisk` outputs would require mental arithmetic or electronic assistance, reducing reliability under time pressure. In contrast, the unit-weighted checklist structure of `AgentScore` was viewed as immediately executable. Participants particularly valued derived features aligned with clinical reasoning patterns (e.g., physiologic ratios such as shock index), which are standard constructs in existing guidelines but are not discoverable by fixed-feature baselines.

**Limitations.**    The sample size ($N = 18$) is modest and precludes subgroup analyses by specialty or experience level. Although participants were blinded to model identity, the structural differences between unit-weighted checklists and integer-weighted scores may have been recognizable. This study assesses face validity and usability preferences; it does not measure actual bedside performance or patient outcomes. Finally, the convenience sampling approach limits generalizability to the broader clinical population.

**Summary.**    Clinicians with substantial experience expressed strong and statistically significant preferences for `AgentScore`-generated unit-weighted checklists over integer-weighted baselines across guideline alignment, bedside usability, and deployment preference. Trust in AI-generated scores increased substantially when large-scale external validation was specified. These findings support the practical deployability of the scoring systems produced by `AgentScore`.

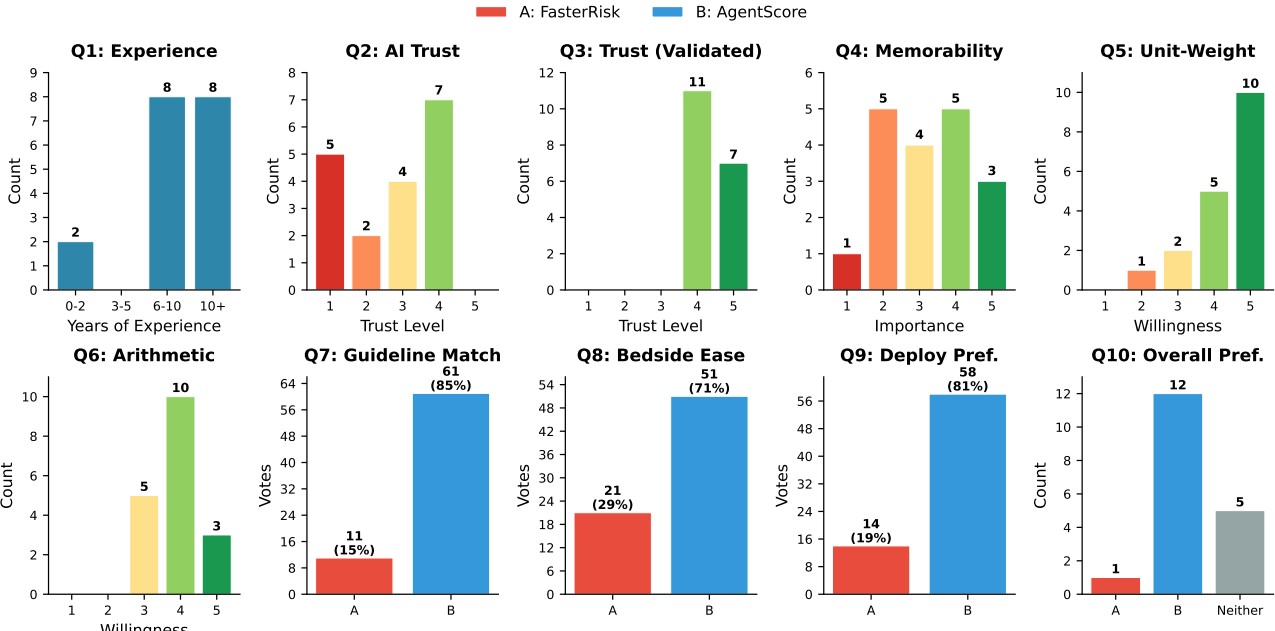

*Figure 8.* **Clinician evaluation of scoring system deployability** ($N = 18$). *Top row:* Experience distribution (Q1) and Likert-scale responses for trust (Q2–Q3) and deployability preferences (Q4–Q6). *Bottom row:* Aggregated pairwise preferences across 4 clinical tasks (Q7–Q9; 72 judgments each) and overall model preference (Q10). Model A: `FasterRisk`; Model B: `AgentScore`.

### E.10. Wall-Clock Time and Cost

We additionally report wall-clock training time for `AgentScore` and all baselines. In absolute terms, `AgentScore` is often slower than standard baselines, primarily due to external API latency associated with language-model queries. Its runtime is nevertheless comparable to optimization-based score-learning methods such as FasterRisk, RiskSLIM, and Optimal Checklists.

For each cross-validation fold, `AgentScore` issues a bounded number of language-model calls. This budget is fixed by pipeline hyperparameters rather than by dataset size. Specifically, we perform up to 100 API calls for candidate rule generation. Each statistically valid candidate rule is then subjected to a single self-consistency check by the language model, resulting in at most an additional 100 calls. Checklist construction involves one initial guideline proposal call, followed by two refinement phases of 10 steps each, yielding at most 21 further calls. In total, the number of API calls per fold is capped at approximately 220.

To make the practical cost transparent, we estimate API cost using the highest-cost model configuration used in our experiments. With input pricing of USD 1.25 per 1M tokens and output pricing of USD 10 per 1M tokens, and observed usage of approximately 2,000 input tokens and 260 output tokens per call, a full fold uses roughly 440,000 input tokens and 57,200 output tokens. This corresponds to an estimated cost of approximately USD 1.12 per fold.

The remaining parts of the pipeline consist of standard CPU-based rule validation, redundancy checking, and checklist assembly rather than GPU-intensive model training. In our setting, these stages can be run without a GPU on a machine with sufficient RAM; for datasets at the scale considered here, this makes experiments feasible on consumer-grade hardware up to approximately 100,000 patients with per-patient feature dimensionality similar to MIMIC.

Training-time compute and deployment-time compute differ fundamentally for `AgentScore`. Computational resources are required only during development. At deployment, the learned $N$-of-$M$ unit-weighted checklist can be evaluated by summing a small number of binary rules, without model-serving infrastructure or language-model inference. Thus, `AgentScore` trades modest additional development-time cost for negligible deployment-time computation, aligning with the constraints of bedside clinical use.

*Table 16.* **Wall-clock training time per fold (seconds).** All PLR variants (RD, RDU, RSRD, Rand, RDP, RDSP) share identical training time and are reported jointly.

| Method | MIMIC AF | MIMIC AKI | MIMIC COPD | MIMIC HF | MIMIC CANCER | MIMIC LUNG | eICU LOS | eICU Vaso |
|---|---|---|---|---|---|---|---|---|
| AgentScore | 3918 | 4096 | 3594 | 2380 | 2244 | 3330 | 3145 | 1382 |
| Optimal Checklist | 3117 | 6610 | 3662 | 7522 | 3604 | 3015 | 3733 | 8700 |
| Decision Tree | 2.9 | 6.0 | 1.9 | 3.7 | 0.4 | 1.3 | 3.3 | 3.2 |
| Logistic | 3.6 | 6.3 | 2.7 | 4.4 | 0.8 | 1.7 | 4.2 | 3.6 |
| FasterRisk | 1360 | 699 | 1045 | 1514 | 55.7 | 588 | 806 | 820 |
| RiskSLIM | 1292 | 193 | 205 | 1176 | 439 | 652 | 1934 | 2942 |
| AutoScore | 156 | 362 | 92 | 153 | 18.3 | 51.9 | 236 | 201 |
| PLR | 61 | 90 | 45.2 | 80 | 8.9 | 32.1 | 151 | 96 |

# F. Algorithmic Details

This section summarizes the `AgentScore` learning procedure and provides concise implementation details to support reproducibility.

## F.1. `AgentScore` Algorithm

---
**Algorithm 2** `AgentScore` Framework
---

**Require:** Training data $(X, y)$, task description $\mathcal{T}$, rule budget $M$
**Require:** Validity threshold $\tau_{\text{rule}}$, redundancy threshold $\delta$
**Ensure:** Unit-weighted checklist $S$ and decision threshold $\tau$
  Split training data into construction set $\mathcal{D}_{\text{con}}$ and validation set $\mathcal{D}_{\text{val}}$
  Construct tool interface $\mathcal{I}$ exposing metadata and aggregate evaluation metrics only
  Initialize rule pool $\mathcal{P} \leftarrow \emptyset$
  {**Phase 1: Rule Pool Generation**}
  **while** generation budget not exhausted **do**
    Propose candidate batch $C$ from grammar $\mathcal{R}$ using **Rule Proposal Agent**
    **for each** proposed rule $r \in C$ **do**
      Compute construction- and validation-split AUROC metrics for $r$ via $\mathcal{I}$ when available
      Compute redundancy metrics using positive-class coverage masks on $\mathcal{D}_{\text{con}}$
      **if** $\text{AUROC}_{\text{val}}(r) \geq \tau_{\text{rule}}$ **and** $\max_{r' \in \mathcal{P}} J_+(r, r') \leq \delta$ **then**
        $plausible \leftarrow$ **Clinical Plausibility Agent**.review($r$)
        **if** $plausible$ **then**
          $\mathcal{P} \leftarrow \mathcal{P} \cup \{r\}$
        **end if**
      **end if**
    **end for**
  **end while**
  {**Phase 2: Score Construction & Refinement**}
  $S \leftarrow$ **Score Construction Agent**.select($\mathcal{P}, M, \mathcal{I}$)
  **while** refinement criteria not met **do**
    $S \leftarrow$ **Score Construction Agent**.refine($S, \mathcal{P}, \mathcal{I}$)
  **end while**
  Select final decision threshold $\tau$ on $\mathcal{D}_{\text{con}}$
  **return** $(S, \tau)$

---

## F.2. Notation, Splits, and Caching

Within each outer cross-validation fold, the training data is further split into a *construction set* $\mathcal{D}_{\text{con}}$ and an internal *validation set* $\mathcal{D}_{\text{val}}$. Candidate-rule evaluation records metrics on both splits when available. Rule acceptance and Phase 2 checklist selection are guided primarily by validation-set performance, falling back to construction-set metrics when validation metrics are unavailable. The decision threshold for the final checklist is selected on $\mathcal{D}_{\text{con}}$, while validation-set AUROC is

reported when available. Held-out test folds are never accessed during rule generation, rule selection, checklist construction, or threshold selection.

We denote by $\mathcal{P} \subset \mathcal{R}$ the pool of retained candidate rules constructed during Phase 1, and by $S \subseteq \mathcal{P}$ the final checklist of size at most $M$ assembled during Phase 2. All scores are unit-weighted by construction.

For each retained rule $r \in \mathcal{P}$, we cache its *positive-class coverage mask* on $\mathcal{D}_{\mathrm{con}}$:

$$(c_r)_i = \mathbf{1}\{r(x_i) = 1 \ \wedge \ y_i = 1\}, \qquad (x_i, y_i) \in \mathcal{D}_{\mathrm{con}},$$

and write $\mathcal{C}_+ = \{c_r : r \in \mathcal{P}\}$. Redundancy is measured using the positive-class Jaccard similarity

$$J_+(r, r') = \frac{|c_r \cap c_{r'}|}{|c_r \cup c_{r'}|}, \qquad r, r' \in \mathcal{P}.$$

### F.3. Tool-Mediated Data Access

All interactions with the dataset are mediated by a fixed tool interface $\mathcal{I}$. The language model never observes raw patient-level values. Instead, it interacts exclusively through deterministic tools that expose: (i) feature names and inferred types; (ii) aggregate statistics computed on training data only (e.g., quantiles, missingness); (iii) evaluation outputs such as AUROC and coverage. No tool returns individual samples, identifiers, or free-form data.

### F.4. Typed Rule Proposal and Validation

Rules are proposed as structured objects drawn from a typed grammar $\mathcal{R}$ supporting thresholds, ranges, derived expressions, temporal summaries, and shallow logical compositions. All proposals must satisfy a strict schema; malformed rules, unknown features, or type violations are deterministically rejected prior to evaluation.

The **Clinical Plausibility Agent** functions as a binary gate applied *only after* statistical screening. It receives the symbolic rule definition and aggregate feature summaries (but no raw data) and returns a binary accept/reject decision based on clinical coherence and interpretability.

### F.5. Deterministic Rule Evaluation and Retention

Each syntactically valid rule is evaluated on the construction split $\mathcal{D}_{\mathrm{con}}$ and retained into the rule pool $\mathcal{P}$ if and only if it satisfies all of the following criteria: (i) its AUROC exceeds a minimum acceptance threshold $\tau_{\mathrm{rule}}$; (ii) its redundancy with previously retained rules, measured by the positive-class Jaccard similarity $J_+$ computed over $\mathcal{D}_{\mathrm{con}}$, is below a fixed threshold $\delta$, unless the rule increases positive-class coverage by at least $\Delta_{\mathrm{min}}$; and (iii) rule-family diversity constraints are satisfied when feasible (i.e., enforcing a minimum mix across rule types such as thresholds, derived rules, and logical compositions when sufficient candidates exist; otherwise skipped).

From the pool of retained rules $\mathcal{P}$, a final checklist $S$ of size at most $m$ is assembled. The decision threshold $\tau$ defining a positive prediction (i.e., $S(\tilde{x}) \geq \tau$) is selected on $\mathcal{D}_{\mathrm{con}}$ according to a specified operating objective. Unless otherwise stated, we use Youden's $J$ as the default threshold criterion, because it provides a standard, assumption-light way to balance sensitivity and specificity when no task-specific utility is specified. The framework is not tied to this choice: PPV, NPV, sensitivity at fixed specificity, balanced accuracy, $F_1$, or cost-sensitive objectives can be used without changing the core method. Importantly, the main AUROC comparisons are threshold-independent. Unless otherwise stated, we use a fixed minimum rule-level acceptance threshold of $\tau_{\mathrm{rule}} = 0.6$ AUROC.

**Computational Complexity.** Conditional on a fixed stream of language-model proposals, the `AgentScore` pipeline is fully deterministic. All rule validation, acceptance, redundancy checks, and checklist assembly are performed via deterministic tools with no access to raw patient-level data. Let $N$ denote the number of samples and $T$ the total number of proposed rules. Rule evaluation scales linearly as $O(T \cdot N)$. Redundancy filtering compares each candidate against the retained pool $\mathcal{P}$ and scales as $O(T \cdot |\mathcal{P}|)$ in the worst case. Coverage masks are stored as compact bitsets of word size $w$, yielding memory usage $O(|\mathcal{P}| \cdot N/w)$ for cached rule coverage, in addition to the data matrix storage. In practice, redundancy filtering dominates runtime, while memory remains modest due to bitset compression.

# G. Agent prompts

This section documents the exact prompting templates used to instantiate the `AgentScore` framework.

---

**Feature Proposal Prompt (AgentScore)**

You are a feature selection agent for clinical risk modeling.

**Task context:**

```
$task_description
```

Propose binary (0/1) clinical features as JSON lines. All rules must follow one of the schemas below.

**Allowed rule types:**

```
numeric_threshold:
{"type":"numeric_threshold","feature":str,"op":">="|">"|"<="|"<","threshold":number}

numeric_range:
{"type":"numeric_range","feature":str,"low":number,"high":number}

categorical_in:
{"type":"categorical_in","feature":str,"in":[category,...]}

binary_true:
{"type":"binary_true","feature":str}

derived_numeric_threshold:
{"type":"derived_numeric_threshold","expr":str,"op":">="|">"|"<="|"<","threshold":number}

count_present:
{"type":"count_present","features":[str,...],"min_count":int}

logical:
{"type":"logical","op":"and"|"or","rules":[<rule>,...]}

percent_change:
{"type":"percent_change","feature_t0":str,"feature_t1":str,
 "pct":number,"op":">="|">"|"<="|"<","direction":"increase"|"decrease"}

zscore_threshold:
{"type":"zscore_threshold","feature":str,"op":">="|">"|"<="|"<","z":number}

quantile_threshold:
{"type":"quantile_threshold","feature":str,"op":">="|">"|"<="|"<","q":float}
```

**Rule diversity guidelines:**

- Use a mix of rule types (thresholds, derived rules, logic).

- Include both high-value ($\geq$) and low-value ($<$) thresholds.

- Prefer rules that capture distinct patient subgroups.

**Clinical interpretability constraints:**

- Prefer features with suffixes `_last`, `_first`, `_min`, `_max`.

- Use round, clinically meaningful thresholds (e.g. HR $\geq$ 100).

- Derived expressions should reflect standard clinical concepts.

**Hard constraints:**

- Output *only* strict JSON.

---

- One rule per line.

- Use only provided variable names.

- Minimum acceptable AUROC when evaluated: `auc_threshold`.

**Available variables:**

`$variable_list`

**Aggregate analysis insights (optional):**

`$analysis_context`

**Tool-derived guidance (optional):**

`$tool_summaries`

Suggest 1–3 candidate rules.

## Clinical Plausibility Review Prompt

You are a clinical expert reviewing a proposed risk prediction rule.
Assess whether the rule is clinically plausible and meaningful.
Consider:

- Whether the direction of risk makes clinical sense.

- Whether thresholds are physiologically reasonable.

- Whether the rule is interpretable and actionable at the bedside.

**Rule under review:**

`$rule_json`

Respond with *only* a JSON object of the form:

`{"plausible": true|false, "reason": "brief explanation"}`

## Score Construction Prompt (N-of-M Checklist)

You are a clinical scoring system designer.

**Task context:**

`$task_description`

You are given a set of binary (0/1) clinical rules.
Construct an interpretable $N$-of-$M$ checklist score with the following properties:

- Each rule contributes exactly 1 point.

- The score equals the count of satisfied rules.

- Predict positive if score $\geq K$ (threshold selected automatically).

- Do not introduce new rules.

**Constraints:**

- Use only the provided rules.

- Maximum number of rules: `max_rules`.

- Output strict JSON with fields:

    - `name`
    - `description`
    - `rules` (list of {`"rule":` `<rule_json>`})

**Candidate rules (with AUROC):**

`$retained_rules_with_auc`

Return *only* the JSON specification.

## Score Refinement Prompt

You are refining an existing clinical checklist score.

**Current score specification:**

`$current_score_json`

**Evaluation metrics:**

`$score_metrics`

**Objective:** Improve discrimination and calibration while preserving interpretability.

**Constraints:**

- Same schema as before.

- Maximum number of rules: `max_rules`.

- All rules are unit-weighted.

- No new rules may be introduced.

Return *only* the updated JSON specification.

