# OpenReview forum: "Automatic Construction of Clinical Scoring Systems with LLM Agents"
_ICML.cc/2026/Conference — ICML 2026 regular_

### Official Review · Reviewer_nKVc · 2026-02-26

**Soundness:** 3
**Presentation:** 3
**Significance:** 4
**Originality:** 4
**Overall Recommendation:** 5
**Confidence:** 4

**Summary:**

Clinicians frequently use clinical scoring systems to assist in making decisions. These systems comprise a set of rules based on clinical measurements. When a rule is satisfied, it gives points. The points from all the rules are summed, and finally, a threshold determines the final decision. These systems and rules are traditionally created manually.

Previous work has attempted to create these systems with machine learning. They use predefined rules and optimize the weight assigned to each rule. AgentScore uses a different approach that can generate new rules. It uses an agent that creates new semantic rules such as physiologic ratios, temporal changes, percentile-based thresholds, and logical compositions. A second agent filters the rules, removing those that are clinically implausible. The final agent iteratively combines the rules into a system, where it cannot include more than M rules. Each rule gives zero or one point, in contrast to previous work, which assigns each rule an integer weight.

They show, across several datasets, that AgentScore outperforms previous automatic rule-based methods, but that unconstrained machine learning methods, such as decision trees, perform better. They show that when they optimize AgentScore on data from one institute, it outperforms manual rules on data from another institute. Finally, they show that clinicians consider the rules generated by AgentScore more clinically useful than those from a previous approach.

**Compliance With Llm Reviewing Policy:**

Affirmed.

**Final Justification:**

"AgentScore: Autoformulation of Deployable Clinical Scoring Systems" is a well-written paper with strong motivation and results. The authors addressed my main concerns, which reinforced my prior assessment. Therefore, I recommend that the paper be accepted.

**Key Questions For Authors:**

1. Why is it important for the rules to be memorizable? Can they not be computed automatically?
2. Do you have evidence for integer-weighted rules being less interpretable? When using unit-weighted rules, are you not encouraging the LLM to create multiple similar rules, instead of fewer rules with higher scores?
3. For the score construction agent, could you have used some of the previous works for adding scores to each rule?

**Limitations:**

Yes

**Strengths And Weaknesses:**

**Strenghts**
* AgentScore is well-motivated, novel, and elegant.
* The paper is well written.
* The experiments are comprehensive.
* The results show that AgentScore outperforms previous approaches.

**Weaknesses**
* I think you are doing yourselves a disservice by not comparing AgentScore with the unconstrained machine-learning approaches in a cross-institutional evaluation (similar to your comparison with the manually created rules). I think that AgentScore would perform better, since the other methods are likely to learn spurious correlations that enhance performance on the institution on which they were trained, but do not generalize to other institutions. A cross-institutional evaluation could reveal these weaknesses in the unconstrained methods.
* You claim that one of the strengths of the rules of AgentScore is that it is more robust towards missing measurements. I do believe this, but you could have shown it by evaluating the methods when randomly dropping measurements.

**Feedback that does not affect the overall score**
* Some of the text in Figure 1 is a bit too small.
* The text is generally well written, but sometimes a bit tough to read. For instance, it is a bit vague what "deployability" exactly means. I suspect this is due to space constraints, so I do not weigh it in my evaluation.

---

> ### Author Rebuttal · Authors · 2026-03-31
>
> We thank the reviewer for the positive assessment of our work, and especially for recognizing the novelty, elegance, and practical importance of AgentScore. We also appreciate the constructive suggestions regarding cross-institutional evaluation, robustness to missing data, and clarification of deployability. We address each point below.
>
> ---
>
> ## Cross-institutional evaluation against unconstrained ML baselines
> We agree that cross-institutional evaluation against unconstrained ML baselines is informative. In our experiments, however, more flexible models such as decision trees still achieve stronger predictive performance than AgentScore even in the external-validation setting. We therefore do not claim that AgentScore outperforms unconstrained ML under cross-institutional shift. This restriction is intentional: the goal of the external-validation analysis is to compare transfer across methods that share the same bedside deployment constraints, rather than against unconstrained predictors designed for a different operating point. For this reason, we focus this comparison on manual clinical rules and the closest learned checklist baseline, Optimal Checklists (see response to Reviewer nRVF).
>
> ## Robustness to missing measurements
> We agree this claim should be supported more directly. Following the reviewer’s suggestion, we added an experiment evaluating robustness under randomly dropped measurements at test time, providing a direct test under partial observability.
>
> Across the two external-validation tasks, both methods degrade with increased missingness, with task-dependent behavior. On ICU Mortality, Optimal Checklists is slightly stronger under full data, while AgentScore remains competitive and both decline as missingness increases. On CF Mortality, AgentScore substantially outperforms Optimal Checklists across all conditions. We therefore adopt a nuanced conclusion: robustness is task-dependent, but AgentScore remains competitive under missingness and compares favorably overall.
>
>
> | Dataset       | Method                  | Missingness | Val AUC       |
> |---------------|--------------------------|-------------|---------------|
> | CF Mortality  | AgentScore               | Full        | 0.601 ± 0.014 |
> | CF Mortality  | AgentScore               | Drop 10%    | 0.591 ± 0.012 |
> | CF Mortality  | AgentScore               | Drop 25%    | 0.576 ± 0.010 |
> | CF Mortality  | Optimal Checklists       | Full        | 0.454 ± 0.113 |
> | CF Mortality  | Optimal Checklists       | Drop 10%    | 0.461 ± 0.091 |
> | CF Mortality  | Optimal Checklists       | Drop 25%    | 0.466 ± 0.069 |
> | ICU Mortality | AgentScore               | Full        | 0.738 ± 0.031 |
> | ICU Mortality | AgentScore               | Drop 10%    | 0.717 ± 0.034 |
> | ICU Mortality | AgentScore               | Drop 25%    | 0.688 ± 0.034 |
> | ICU Mortality | Optimal Checklists       | Full        | 0.759 ± 0.008 |
> | ICU Mortality | Optimal Checklists       | Drop 10%    | 0.746 ± 0.010 |
> | ICU Mortality | Optimal Checklists       | Drop 25%    | 0.710 ± 0.008 |
>
>
> ## Rule memorability and unit weights
>
> We agree that deployability should be clarified. Our claim is not that alternative scoring methods are less interpretable, but that deployability imposes stricter requirements. In our setting, deployability refers to scoring systems that are compact, unit-weighted, auditable, and usable at the bedside without specialized infrastructure. This is formalized in Definition 2.1 through parsimony, interpretability, memorability, operational deployability, and predictive adequacy.
>
> This distinction is particularly relevant in time-critical settings (e.g., ICU triage), where one may not assume access to a precomputation pipeline. While precomputation is possible, once reliable software-mediated inference is assumed, the advantage of checklist-style scores diminishes relative to other interpretable models (e.g., logistic regression or decision trees). Our focus is therefore on scores that remain natively usable and auditable.
>
> We also note that unit weighting does not encourage duplication of similar rules, as redundancy is explicitly controlled via overlap filtering during validation and a bounded checklist size.
>
> ## Presentation
> We have revised Figure 1 and expanded the explanation of deployability in the main text accordingly.
>
>
>
> ## Score construction agent
> We now also report results combining AgentScore-derived rules with FasterRisk (see response to Reviewer X3fp). However, AgentScore’s full pipeline still matches or outperforms these variants across datasets. This suggests that while rule quality is important, the proposal–validation–assembly procedure provides additional gains beyond what can be achieved by standard score-construction methods.
>
>
> ---
> We thank the reviewer again for the thoughtful and constructive feedback. We believe these additions and clarifications further strengthen the paper.

---

> > ### Author Rebuttal · Reviewer_nKVc · 2026-04-01
> >
> > Dear authors, thank you for an excellent paper and for clear and thorough responses. I stand by my score at 5, which is a very high score.

---

> > > ### Author Response · Authors · 2026-04-05
> > >
> > > We thank the reviewer for their careful consideration of our rebuttal and for the constructive feedback throughout the review process. We greatly appreciate the positive assessment of our work and are glad that our responses addressed the concerns. We are grateful for the reviewer’s suggestions, which have helped strengthen both the empirical evaluation and the clarity of the paper.

---

### Official Review · Reviewer_nRVF · 2026-03-06

**Soundness:** 3
**Presentation:** 3
**Significance:** 2
**Originality:** 2
**Overall Recommendation:** 4
**Confidence:** 2

**Summary:**

The paper proposes a novel workflow for creating interpretable predictive checklists for clinical prediction. The workflow consists of several stages. First, the method uses a language model to generate clinically plausible rules that passes minimal predictive targets on the private dataset. Second, the method uses a language model in an iterative way to compose rulesets, each evaluated on the private datasets. The paper evaluates the method on a set of realistic clinical prediction tasks, performs an ablation study, and performs a clinical validation, showing the superiority of the models to other interpretable models.

**Compliance With Llm Reviewing Policy:**

Affirmed.

**Final Justification:**

I feel quite conflicted about the outcome of the discussion. On the one hand, I absolutely agree with the other reviews that the paper is high-quality in its evaluation and algorithm design. On the other hand, I do believe that a major revision is more fair to properly position the work within the literature. I tried to emulate a journal-style revision by asking the authors for specific revised wording in their 2nd response. Because of such a limited format and timeline, I think the resulting revision is more of a patch rather than a faithful re-framing. Concretely, the revision mostly kept the original wording and mostly added additional qualifiers. I believe a more fair revision would center the contribution of this paper around feature search/selection – which I believe is the conceptual delta – and not the "deployable models" as a whole. As novelty is one of the core values at NeurIPS, I feel strongly about doing justice to the prior work. Trying to balance the concerns, however, I think the new method and its strong performance somewhat outweigh the concerns for fairness to prior work, so I would narrowly lean towards acceptance.

**Key Questions For Authors:**

–

**Limitations:**

The limitations are adequately addressed. I would add more on disaggregated performance evaluation across salient demographic groups.

**Strengths And Weaknesses:**

The **idea of learning predictive checklists has been introduced in a NeurIPS 2021 paper by [Zhang et al.](https://arxiv.org/abs/2112.01020),** with a specially optimized integer-programming method that automatically selects and discretizes features to learn optimal checklists. That paper has exactly the same goal as the present manuscript, and solves it well. This substantially reduces the contributions in the present manuscript (in particular, the "conceptual contribution" in the list of contributions). It is also unclear whether the optimal checklists can achieve the performance of AgentScore in a substantially cheaper way without using language models, what is the relative performance difference, and whether AgentScore can incorporate the optimal checklist solvers as part of its operation.

The manuscript should incorporate the following changes:
- Re-position the introduction and related work to make it clear that the method does not present a new concept of learning checklists, and instead proposes a novel LLM-assisted method to generate them.
- Incorporates optimal checklists as part of the benchmark suite.
- Considers options for integrating the parts of the existing algorithms for learning optimal checklists as part of AgentScore.

These changes alone would constitute a major revision that requires another round of review. Because of this, I recommend such a low score. This is very unfortunate, as otherwise the paper would have been very good: well-written, for the most part clear, with an interesting and potentially useful method, and an extensive, realistic, evaluation in realistic tasks.

---

Beyond this major problem, there are several other issues:
- _Unclear presentation of the method._ The description of the method in the main body is not clear enough. I believe the pseudocode, or its simplified version should be in the main body; otherwise it's not possible to reproduce the method from the textual description alone.
- _Ignored privacy leakage._ Although the exact algorithm is unclear, L298 suggests that the performance metrics might be released to the language model API. This leakage might be minimal, but prior work, e.g., synthetic data generation from APIs ([Lin et al.](https://openreview.net/forum?id=YEhQs8POIo), ICLR 2024), does take care of privacy risk in exactly this setting. This needs to be at least explicitly discussed and acknowledged, with pointers on how to privatize the procedure.
- The choice of Youden's J as the main selection metric is unclear. Why not, e.g., PPV at a realistic prevalance of a condition being predicted?

---

> ### Author Rebuttal · Authors · 2026-03-31
>
> We thank the reviewer for the careful assessment of our work and for highlighting the connection to Zhang et al.
>
> ---
>
> ## Revised positioning
> We agree that Zhang et al. introduced checklist learning, and checklist-style scores have long existed in medicine. Our contribution is not checklist learning per se, but learning deployable clinical checklists from raw clinical variables via a clinically motivated rule language, rather than selecting over fixed pre-binned or manually specified features.
>
>
> We will make this distinction much clearer in the revision. Specifically, we will:
>
> 1. Explicitly acknowledge Zhang et al. in the Introduction and narrow our novelty claim accordingly;
> 2. Cite Zhang et al. in the Deployable Clinical Scoring Systems section when introducing checklist structure, while clarifying that our formulation adds a clinically motivated rule language.
> 3. Add Optimal Checklists to Related Work and Table 2 as a direct checklist-learning baseline: a unit-weighted N-of-M checklist over fixed binned/manual candidate features, optimized with MIP but without automatic derived-rule construction.
>
> That said, Zhang et al. does not fully address our stricter notion of deployability: it operates over preprocessed features, and the resulting rules can involve thresholds (e.g., AST ≥ 162.6 IU/L) that are less suitable for routine bedside use.
>
> ## Optimal checklists in the benchmark suite
>
> Empirically, Optimal Checklists is competitive on some tasks but underperforms overall:
>
> | Dataset                  |      AgentScore | Optimal Checklists |
> | ------------------------ | --------------: | -----------------: |
> | MIMIC AF                 | **0.81 ± 0.01** |        0.59 ± 0.16 |
> | MIMIC AKI                | **0.79 ± 0.02** |      0.68 ± 0.00 |
> | MIMIC COPD        |     0.63 ± 0.03 |   **0.66 ± 0.02** |
> | eICU LOS                 | **0.67 ± 0.00** |      0.59 ± 0.00 |
> | eICU Vasopressor         | **0.76 ± 0.00** |       0.54 ± 0.01 |
> | MIMIC HF    | **0.79 ± 0.01** |       0.56 ± 0.03 |
> | MIMIC Cancer | 0.59 ± 0.01 |       **0.62 ± 0.02** |
> | MIMIC Lung|     0.63 ± 0.00 |  **0.67 ± 0.01** |
> | **Mean**  | **0.71 ± 0.08** |    **0.61 ± 0.05** |
>
>
> ## Integrating optimal checklists with AgentScore
> A natural hybrid is to use AgentScore for rule generation and a classical optimizer for final selection over the retained rule pool. We tested both greedy and MIP-based (using Optimal Checklists) selection under identical checklist constraints (see response to Reviewer X3fp). Greedy performs poorly, and MIP improves on it but still underperforms AgentScore, particularly on harder tasks. Thus, even on the same validated rule pool, AgentScore’s LLM-based assembly step performs better on held-out evaluation than the classical exact-selection variants we tested.
>
> ## Pseudocode inclusion
> We will move a simplified pseudocode summary into the main paper to improve clarity and reproducibility.
>
> ## Privacy leakage
>
> Our framework reduces exposure by restricting the LLM to aggregate statistics rather than raw patient-level data. However, this is not a formal privacy guarantee when using external API-based models. This risk is reduced when using locally hosted open-source models.
>
> We also ran a membership-inference-style privacy audit under three attacker settings, where “access” means what information the attacker is allowed to observe:
>
> Transcript only: the agent’s tool-interaction history only
>
> Final only: the final checklist only
>
> Strong: both the transcript and the final checklist
>
> Across the 8 MIMIC and eICU datasets, attack performance was close to chance:
>
>
> | Attack setting   |      Mean AUC | Mean advantage | TPR @ 1% FPR |
> | ---------------  | ------------: | -------------: | -----------: |
> | Strong                   | 0.509 ± 0.044 |         +0.018 |        0.055 |
> | Final only      |          0.507 ± 0.056 |         +0.015 |        0.048 |
> | Transcript only |         0.501 ± 0.021 |         +0.002 |        0.016 |
>
>
>
> ## Threshold selection
> We use Youden’s J as a default threshold criterion because it is a standard, assumption-light way to balance sensitivity and specificity when no task-specific utility is specified. We agree it is not always optimal; depending on the clinical setting, PPV, NPV, sensitivity at fixed specificity, or cost-sensitive criteria may be more appropriate. The framework is not tied to Youden’s J, and alternative threshold criteria can be used without changing the core method. Our main AUROC comparisons are threshold-independent.
>
> ## Limitations
>
> We agree that subgroup fairness deserves clearer discussion. Our current framework does not provide statistical guarantees across subgroups, and like any data-driven model it may reflect biases present in the training data.
>
>
>
> ---
> We thank the reviewer again for these comments. We believe the clarified positioning, added Optimal Checklists benchmark, and privacy audit materially strengthen the paper. We'd be happy to engage in further discussion.

---

> > ### Author Rebuttal · Reviewer_nRVF · 2026-04-02
> >
> > *Update*. Apologies, I incorrectly posted the response in a separate note that was only readable by chairs/other reviewers. Updating the acknowldgement here.
> >
> > _1. Revisions_
> >
> > > Specifically, we will [address the three points]
> >
> > In order to make a recommendation, I would like to see the exact wording of the revisions.
> >
> > Please detail the proposed revisions of the relevant parts of the paper: the "gap" paragraph, the revised "contribution" paragraph/box, any other relevant portions of the introduction, related work, conclusions. The most important part is the updated contributions.
> >
> > _2. Details_
> >
> > - What were the input feature representations in the new Optimal Checklists experiments?
> > - How exactly did you run the MIAs? Which algorithms and adversary's data settings?

---

> > > ### Author Response · Authors · 2026-04-05
> > >
> > > We thank the reviewer for the detailed follow-up. Below we provide the exact revisions and clarifications requested.
> > >
> > >
> > > ### Intro
> > > We will cite Zhang et al. in line 20 when introducing ML models optimised for bedside execution.
> > >
> > >
> > > We will make the following changes to the gap and contribution sections:
> > > ### Gap
> > > `Conversely, existing integer risk scores and checklist learning methods largely optimize over a fixed, pre-constructed feature matrix (cite FasterRisk and Optimal Checklists) and therefore do not address the setting where clinically meaningful derived rules must be constructed from raw variables.`
> > >
> > > ### Conceptual
> > >
> > > `We formalize deployable clinical checklist learning as a joint combinatorial optimization problem over rule construction and subset selection, rather than checklist selection over fixed pre-constructed features, within a constrained model class of unit-weighted checklists. The rule language supports temporal patterns, physiologic ratios, and shallow compositions, explicitly encoding bedside cognitive and operational constraints.`
> > >
> > > ### Empirical
> > > `AgentScore matches or exceeds state-of-the-art integer score and checklist learning baselines despite operating under stricter structural constraints.`
> > >
> > >
> > > ### Formalism
> > > We will cite Optimal Checklists in line 63 when introducing the checklist structure.
> > >
> > > We will also add the following sentence in line 97:
> > > `Furthermore, prior work has shown that automatically constructed checklist models can achieve performance comparable to more flexible models when operating over fixed feature representations (cite Optimal Checklists).`
> > >
> > >
> > > ### Related work
> > > Table 2 will include Optimal Checklists as:
> > >
> > > | Method | Score form | Weights | Rule language | Search / construction | Derived rule construction | Unit-weighted unordered checklist |
> > > |--------|-----------|--------|---------------|-----------------------|---------------------------|----------------------------------|
> > > | Optimal Checklists | N-of-M checklist | unit (0/1) | binned thresholds / fixed binaries | MIP over rule selection | × | ✓ |
> > >
> > > We will also add the following in line 176:
> > >
> > > `Relatedly, prior work has also considered direct learning of unit-weighted checklists. In particular, Optimal Checklists (cite Optimal checklists) formulates checklist learning as mixed-integer optimization over fixed candidate features, yielding unit-weighted N-of-M checklists without coefficient arithmetic. However, these methods assume a fixed candidate feature space and do not address the joint problem of constructing and selecting derived clinical rules.`
> > >
> > >
> > >
> > > ### Discussion
> > > We will edit the phrasing in line 419:
> > > `From this perspective, clinical scoring systems are not a legacy artifact to be replaced, but a deliberately constrained hypothesis class optimized for deployment, where strong performance depends not only on selecting a sparse checklist, but also on constructing clinically meaningful rules from raw variables.`
> > >
> > >
> > >
> > > We will also add the following sentence in the conclusion:
> > > `More broadly, our results suggest that learning deployable scores benefits from treating rule construction and checklist assembly as a joint optimization problem, rather than learning an optimal checklist over a fixed pre-constructed feature space.`
> > >
> > > ## Implementation details
> > > ### Optimal checklists
> > > Optimal Checklists was evaluated under three input settings:
> > > (i) raw clinical variables (see previous response),
> > > (ii) the validated rule pool produced by AgentScore (see Response to Reviewer X3fp).
> > > (iii) the raw clinical variables and the validated rule pool produced by AgentScore (Table below)
> > >
> > > In all cases, the same train/test splits were used as for AgentScore. In the raw-variable setting, Optimal Checklists operates on the same precomputed deployable feature representation used by non-agent baselines (i.e., static summaries and binned clinical variables), but without any LLM-derived rules. We used the official implementation provided by the original authors for raw-feature processing and training.
> > >
> > > | Dataset             |           AUROC |
> > > | ----------| ----------: |
> > > | MIMIC AKI               | 0.78 ± 0.04 |
> > > | MIMIC AF                 | 0.72 ± 0.07 |
> > > | MIMIC HF   | 0.68 ± 0.10 |
> > > | eICU LOS                 | 0.66 ± 0.00 |
> > > | MIMIC LUNG   | 0.62 ± 0.01 |
> > > | MIMIC COPD       | 0.60 ± 0.05 |
> > > | MIMIC CANCER | 0.59 ± 0.01 |
> > > | eICU Vasopressor         | 0.50 ± 0.00 |
> > > | Mean        | 0.64 ± 0.04 |
> > >
> > >
> > >
> > > ### MIA
> > > For each dataset, we run a membership-leakage audit using a no-LLM oracle that simulates AgentScore-style rule selection on paired held-out evaluation worlds differing by one target-vs-donor substitution. From each run, we build attack features from either the interaction transcript (transcript_only), the final selected rules (final_only), or both (strong). We then train a balanced logistic-regression attacker on generated attack samples using a grouped 80/20 split by target identity, and report attack AUC, advantage (2*AUC - 1), and TPR at 1% and 5% FPR.

---

### Official Review · Reviewer_X3fp · 2026-03-12

**Soundness:** 3
**Presentation:** 3
**Significance:** 4
**Originality:** 3
**Overall Recommendation:** 5
**Confidence:** 3

**Summary:**

Existing score-learning methods learn integer-weighted models with non-unit coefficients, but the format clinicians actually employ at the bedside is the unit-weighted unordered checklist, where every rule contributes +1 and the decision reduces to counting. No prior method directly optimizes over this model class for the ease of bedside deployment. This paper fills the gap: it formulates checklist learning as a combinatorial optimization problem over a discrete rule space generated from a clinically motivated grammar (supporting ratios, temporal patterns, and shallow logical compositions), subject to hard constraints on checklist size and rule depth. Then, it tries to solve the problem via an iterative search-and-validate framework: LLMs navigate the combinatorial rule space while a deterministic verification loop enforces statistical validity. On 8 MIMIC-IV/eICU tasks, AgentScore achieves mean AUROC 0.71 versus 0.66 (RiskSLIM) and 0.68 (FasterRisk), outperforms SOFA/SAPS-I and CF guidelines on external validation, and is strongly preferred by clinicians (n=18, blinded) in a pairwise preference survey on guideline alignment and deployment suitability.

**Compliance With Llm Reviewing Policy:**

Affirmed.

**Final Justification:**

My concerns have been addressed during the rebuttal. Though missing very critical prior work as the other reviewer points out, the additional experiments have added it for comparison. I expect more details regarding the use and tuning of MIP formulation (objective, constraints) in the revision. Though the performance improvement seems modest, I appreciate the work focus on the deployable aspect than the interpretable aspect with the clinicians preference study. Therefore, I raise my score to 5.

**Key Questions For Authors:**

I think this is a strong and timely work, and a few targeted experiments would help me change my evaluation of the paper. 1) Adding a baseline that gives RiskSLIM or FasterRisk access to the same derived features AgentScore generates. This would help readers understand where the gains are really coming from. 2) Add a greedy selection or MIP baseline (if resources available) on the retained rule pool would clarify the role of the LLM at the assembly step. 3) It would also help to apply the same significance testing to the ablation study.

A couple of smaller clarifications: what training data the baselines use relative to AgentScore's rule construction/validation split? Finally, a brief discussion of how AgentScore relates to the growing literature on LLM-guided combinatorial search would help readers appreciate both its novelty and its generalisability.

**Limitations:**

yes

**Strengths And Weaknesses:**

The distinction between "interpretable" and "deployable" is precise and practically important. Formalizing unit-weighted unordered checklists as a constrained model class is a concrete contribution that prior score-learning work does not make. The (iterative) search-and-validate approach is well-motivated: LLMs are capable proposal generators over a vast combinatorial rule space, but as stochastic models they will inevitably produce invalid or redundant candidates. The deterministic validation layer (e.g., a minimum discrimination threshold and Jaccard-based redundancy checks) is therefore not just a design choice but a necessity, guiding the search toward statistical validity. The ablation study makes this concrete: unconstrained LLM generation alone collapses to 0.59 AUROC. Most importantly, including a blinded, pre-specified clinician preference study alongside predictive metrics meaningfully strengthens the case for practical deployment value.

The authors have not discussed the broader literature on LLMs as proposal/candidates generators over combinatorial search spaces and serve an a critical component within a iterative/evolving framework with verifiers/oracles. I believe that the proposed method shares the same spirit with this line of work, and situating itself within this literature and clarifying what is specific to clinical deployment versus what is a general instantiation of LLM-guided combinatorial search would help readers assess its novelty. Despite the aggregate significance shown in Table 3, the large variance across tasks suggests the gains are not uniform, but at least it is backed up significance tests so that the readers know AgentScore is better than FasterRisk and RiskSLIM. However, the ablations in Table 4 has the similar presentation but without significance tests. Reporting effect sizes and confidence intervals for the ablations, as done for the main comparison with baselines, would give a better justification of the component design. Furthermore, I found it difficult to apprehend if the performance gain comes from LLM-guided search or the richer feature grammar or both. Though AgentScore auto-generate features, which is part of its contribution as compared to manual feature engineering, no baselines have access to these features. I think a baseline given the same pre-computed derived features would quantify how much of the AUROC gain comes from the search method versus the richer feature set. Lastly, the justification for using an LLM agent over classical solvers at the assembly step is weak. The assembly step operates on the retained pool, which is a smaller space, and commercial and open-source MIP solvers might be able to solve it. These solvers can also handle constrained combinatorial optimization problems such as enforcing diversity as a constraint (if the concerns is semantic diversity) in a MIP, yet it is never compared against.

---

> ### Author Rebuttal · Authors · 2026-03-31
>
> We thank the reviewer for the thoughtful and constructive review. We address each point below with additional experiments and analyses.
>
> ---
>
> ## Baselines with access to the same derived features
> To disentangle feature construction from search, we evaluated classical score-learning methods on the same derived rule pool generated by AgentScore.
>
> This isolates two effects: (i) the value of clinically structured derived rules, and (ii) the effectiveness of the proposal--validation--assembly pipeline given a fixed representation.
>
> Across datasets, AgentScore matches or outperforms FasterRisk under this setting, indicating that gains are not solely attributable to feature construction but also to the search and assembly procedure.
>
>
> | Method                                      | MIMIC AF        | MIMIC COPD      | MIMIC HF        | MIMIC AKI       | MIMIC Cancer    | MIMIC Lung      | eICU LOS        | eICU Vaso       | Mean            |
> | ------------------------------------------- | --------------- | --------------- | --------------- | --------------- | --------------- | --------------- | --------------- | --------------- | --------------- |
> | **AgentScore (full pipeline)**              | **0.80 ± 0.01** | 0.63 ± 0.03     | **0.79 ± 0.01** | **0.79 ± 0.02** | **0.59 ± 0.01** | **0.63 ± 0.00** | **0.67 ± 0.00** | **0.76 ± 0.00** | **0.71 ± 0.08** |
> | AgentScore + FasterRisk                     | 0.79 ± 0.02     | 0.63 ± 0.00     | 0.77 ± 0.03     | 0.77 ± 0.02     | 0.58 ± 0.01     | 0.58 ± 0.01     | 0.65 ± 0.01     | 0.74 ± 0.01     | 0.69 ± 0.08     |
> | AgentScore + MIP (Optimal Checklist Solver) | 0.76 ± 0.03     | **0.64 ± 0.02** | 0.69 ± 0.08     | 0.75 ± 0.04     | **0.59 ± 0.01** | 0.59 ± 0.01     | 0.66 ± 0.00     | 0.50 ± 0.08     | 0.65 ± 0.09     |
> | AgentScore + Greedy Selection               | 0.56 ± 0.07     | 0.52 ± 0.05     | 0.55 ± 0.06     | 0.61 ± 0.06     | 0.50 ± 0.00     | 0.50 ± 0.00     | 0.50 ± 0.00     | 0.50 ± 0.08     | 0.53 ± 0.05     |
>
>
>
>
> ## Greedy / MIP baseline on the retained rule pool
> We agree that classical optimizers become viable once the rule pool is reduced, and we evaluated both greedy selection and MIP-based optimization under identical checklist constraints.
>
> Greedy selection performs poorly, often selecting highly overlapping rules due to lack of redundancy control, leading to near-random performance (AUC ≈ 0.5) on several datasets.
> MIP-based optimization improves over greedy selection, but still underperforms AgentScore, particularly on more complex tasks.
>
> Notably, even when optimizing the same objective on the same candidate pool, the LLM-based assembly step yields better generalization. These results suggest that semantically guided assembly may provide a useful regularization effect by favoring clinically diverse and coherent rule sets, which are difficult to encode using purely combinatorial constraints.
>
>
>
> ## Significance testing of ablations
> We have extended the ablation analysis to include statistical testing. Across all comparisons, the full AgentScore pipeline significantly outperforms ablated variants ($p < 0.05$, Wilcoxon signed-rank test; Holm-corrected).
>
> Effect sizes (Cohen’s $d_z$) range from ~1.0 for milder ablations to ~2.8 for removing deterministic validation, confirming that each component contributes meaningfully, with the largest degradation arising when validation is removed.
>
> ## Clarification on Training Data Splits
> To address the question regarding data splits: both AgentScore and the baselines are evaluated using the exact same overarching training and test sets to ensure a fair comparison. Within the training set, AgentScore internally holds out a 20% validation split specifically for its deterministic rule validation loop. The baselines utilize the training data according to their standard respective optimization procedures.
>
>
> ## Extended related work
>
> AgentScore is related in spirit to recent work that uses LLMs to propose structured candidates within search-and-verify or search-and-optimize loops, including FunSearch, OPRO, LLM-SR, and LLM-based evolutionary optimization [1-4].  At the same time, our setting differs in a crucial way: the hypothesis class is not an unconstrained search space, but one explicitly defined by clinical deployment requirements rather than purely predictive optimization.
>
>
> [1]. LLM-SR: Scientific Equation Discovery via Programming with Large Language Models (Shojaee et al. ICLR 2026)
>
> [2]. Mathematical discoveries from program search with large language models (Romera-Paredes et al. Nature 2023)
>
> [3]. Large Language Models as Optimizers (Yang et al. ICLR 2024)
>
> [4]. Large Language Models as Evolutionary Optimizers (Liu et al. IEEE Evol. Comp. 2024)
>
> ---
>
> We thank the reviewer again for the constructive feedback. We believe these additional experiments directly address the reviewer’s concerns and clarify the roles of feature construction, search, and assembly. We’d be happy to engage in further discussions.

---

> > ### Author Rebuttal · Reviewer_X3fp · 2026-04-02
> >
> > I appreciate the additional ablation of FasterRisk and MIP with access to the same derived features. As you point out that there are some near-random entries (AUC ≈ 0.5) and it would be better discuss their failure modes as they only happen to certain datasets. Please incorporate the positioning with LLM-guided combinatorial search in the revision. Based on the full resolution, I will raise my score.

---

> > > ### Author Response · Authors · 2026-04-05
> > >
> > > We thank the reviewer for the positive assessment of the additional experiments and for raising their score. We will incorporate a more detailed discussion of the observed failure modes of greedy and MIP-based approaches (e.g., near-random performance on certain datasets) in the appendix to clarify when and why these methods underperform.
> > >
> > > **Illustrative failure mode.**
> > > Let $A, B \sim \text{Bernoulli}(0.5)$ be independent binary clinical factors, and define
> > > $$
> > > Y = \mathbb{1}[A + B \ge 2] = \mathbb{1}[A=1,\,B=1].
> > > $$
> > > Thus, positives are patients in whom *both* risk factors are active, while all others are negative. Correct prediction therefore requires complementary coverage of *both* factors.
> > >
> > > Suppose the candidate rule pool contains multiple variants of $A$ (e.g., indirect metrics for respiratory failure) and fewer rules for $B$ (e.g., signs of infection). If rules corresponding to $A$ exhibit slightly higher marginal validation performance, a **greedy or otherwise myopic selector** may choose a checklist $S = (r_A^{(1)}, r_A^{(2)})$ consisting only of redundant $A$-type rules. Such a checklist cannot distinguish $(A=1,B=0)$ from $(A=1,B=1)$, since both receive the same score even though the former is negative and the latter is positive. This illustrates why greedy selection can fail even over the same retained rule pool: redundant high-marginal rules can crowd out complementary ones that are necessary to capture the target structure.
> > >
> > > For **MIP-based selection**, the issue is more subtle. In principle, an exact solver with a fully appropriate objective and constraints could recover the complementary pair. However, under the same reduced rule pool and operational constraints used in our experiments, MIP still optimizes only the specified objective over the available candidates and does not explicitly encode semantic complementarity across physiological domains. As a result, it can still favor statistically similar or redundant rules when complementarity is not well captured by the optimization formulation. We will clarify this distinction in the revision and discuss the dataset-specific failure modes in more detail.
> > >
> > > We will also incorporate the suggested positioning of AgentScore within the broader literature on LLM-guided combinatorial search in the revision.

---

### Official Review · Reviewer_yzL3 · 2026-03-13

**Soundness:** 3
**Presentation:** 4
**Significance:** 3
**Originality:** 3
**Overall Recommendation:** 5
**Confidence:** 2

**Summary:**

The paper introduces AgentScore, which combines LLM-guided rule proposal for clinical guidelines and validation to learn deployable scorers. Starting from a grammar of rules, AgentScore generates candidate rules, which get filtered deterministically and then another agent assembles a final checklist from the validated rules. The paper aims to formulate clinical deployability as a first-class model constraint. They evaluate their method on 8 clinical prediction tasks and 2 external validation tasks, and outperform existing baselines.

**Compliance With Llm Reviewing Policy:**

Affirmed.

**Final Justification:**

The paper's formulation is principled. I stick to accept.

**Key Questions For Authors:**

Mentioned along with weaknesses.

**Limitations:**

Mentioned along with weaknesses.

**Strengths And Weaknesses:**

Strengths:
- The authors provide a principled formulation to tackle the bottleneck of translating models into clinic by treating deployability as a modeling constraint. This provides a new direction of research to incorporate clinical guidelines into practice through LLMs
- The separation of generated proposals to deterministic validity check makes sense, as the rule-selection is data-grounded and since the LM doesn't see raw patient data, it is privacy-preserving
- Addition of human eval to usability provides strong evidence that the system might be favored in the clinic

Weaknesses/Questions:
- How is $\tau_{\text{rule}}$ chosen? Is it separate for each rule or can it be learnable?
- I would be interested to see a cost-analysis for the system. Can it be added? (such as how many times the LLM was called)?
- Did the authors try to compare to search heuristic baselines over the combinatorial search space? It might be nice to see if the LLM's contributions outperform such heuristics
- Maybe for readability, it would be useful to remove some of the PLR baselines from the main (since most of them have similar performance)

I think the paper motivates a strong idea, and provides a systematic framework for constructing these deployable systems with emphasis on checklist-style scores. I recommend accept.

---

> ### Author Rebuttal · Authors · 2026-03-31
>
> We thank the reviewer for the positive assessment and for highlighting the importance of clinical deployability. We also appreciate the suggestions on $\tau_{\text{rule}}$, cost transparency and heuristic baselines.
>
> ---
>
> ## Choice of $\tau_{\text{rule}}$
> We use $\tau_{\text{rule}} = 0.6$ as a minimum rule-level AUC threshold to filter clearly non-informative candidates while retaining sufficient flexibility for downstream assembly. This acts as a screening hyperparameter that encodes the prior that each retained rule should have at least modest standalone predictive value.
>
> If set too low, the filter admits many weak or redundant rules; if too high, it over-restricts the pool and harms assembly. Across datasets, $\tau_{\text{rule}}=0.6$ provided a robust screening level that removed weak candidates without materially shrinking the validated pool needed for downstream checklist assembly. In practice, for most datasets, substantially more rules passed this threshold than were ultimately needed for the final checklist, indicating that $\tau_{\text{rule}}$ served primarily as a quality filter rather than a binding constraint on assembly.
>
> We also explored dataset-adaptive thresholds based on simple univariate signal estimates, but observed no consistent improvement over the fixed threshold. In some cases, higher thresholds over-pruned complementary rules, leading to slightly worse performance. We will clarify this role of $\tau_{\text{rule}}$ in the revision.
>
>
>
> ## Cost analysis / number of LLM calls.
> We already report the number of LLM calls in Appendix E.7. AgentScore makes at most 220 LLM calls per fold under the hyperparameter setting used in our experiments; this call budget is fixed by the pipeline hyperparameters rather than by dataset size.
>
> To make the practical cost more transparent, we will add an explicit estimate based on the highest-cost model configuration used in our experiments (GPT-5). Using current API pricing (input: USD 1.25 per 1M tokens, output: USD 10 per 1M tokens), and the observed token usage in our pipeline, each call uses approximately 2,000 input tokens and 260 output tokens. Across 220 calls, this corresponds to roughly 440,000 input tokens and 57,200 output tokens, yielding an estimated cost of approximately **USD 1.12 per fold**.
>
> More broadly, we believe this also strengthens the practical accessibility of the method. The LLM component is lightweight in absolute cost, and the remaining parts of the pipeline are standard CPU-based validation and assembly procedures rather than GPU-intensive training. In our setting, these stages can be run without a GPU on a machine with sufficient RAM; for most datasets of the scale considered here, this means that experiments are feasible on consumer-grade hardware up to roughly **100,000 patients** with per-patient feature dimensionality similar to MIMIC. We will clarify this point in the revision, as accessibility and ease of deployment are important aspects of the proposed framework.
>
> ## Comparison to heuristic search baselines.
> We agree that isolating the contribution of LLM-guided proposal generation is important. As discussed in Appendix E.1, the full constrained search space is combinatorial and cannot be exhaustively enumerated in our setting. More importantly, even strong classical solvers operating over raw variables and thresholds do not guarantee that the resulting rules correspond to clinically meaningful constructs; they may recover statistically useful but semantically arbitrary combinations.
>
> AgentScore addresses this by restricting proposals to a clinically motivated rule grammar and using the LLM to generate semantically structured candidates, so its contribution is not only search efficiency but also biasing exploration toward clinically interpretable rule forms.
>
> To directly address the reviewer’s suggestion, we have added two new comparisons: (1) AgentScore against FasterRisk (see response to Reviewer X3fp), where FasterRisk is given access to the LLM-generated features, and (2) a MIP solver for unit checklists operating only on the raw patient features (see response to Reviewer nRVF). These results suggest that a substantial part of the gain comes from richer semantic rule construction, while also showing that classical optimization over raw features alone is insufficient to recover the same performance.
>
>
> ## Readability
> We agree. Retaining only the strongest PLR variant in the main table reduces visual noise and makes the comparison structure clearer, the key contrast is between AgentScore and the best-performing integer-score baselines, not across PLR rounding strategies. In the revision, we will move the remaining variants to the appendix and add the Optimal Checklists baseline requested by Reviewer nRVF.
>
>
> ---
> We thank the reviewer again for the supportive and constructive feedback. We hope these clarifications and additions address the remaining concerns. We’d be happy to engage in further discussions.

---

> > ### Author Rebuttal · Reviewer_yzL3 · 2026-04-03
> >
> > The reviewers have answered my concerns and I maintain my score.

---

> > > ### Author Response · Authors · 2026-04-05
> > >
> > > We thank the reviewer for their careful consideration of our rebuttal and for the constructive feedback throughout the review process. We appreciate the positive assessment that the concerns have been addressed, and we are grateful for the role this feedback has played in strengthening the paper.

---

### Decision · Program_Chairs · 2026-04-30

**Decision:**

Accept (regular)

**Comment:**

The paper introduces AgentScore, a system for LLM-guided development of clinically deployable risk scoring systems in the form of uniformly weighted checklists.

The manuscript puts forward three claimed contributions: 1) Formalizing deployable clinical scoring systems as a constrained model class defined by unit-weighted checklists (...), 2) A framework that bridges LLMs and discrete optimization, 3) Matching or
exceeding state-of-the-art score-learning baselines under stricter structural constraints.

Contribution 1) is not novel as originally phrased, as pointed out by Reviewer nRVF. The authors addressed this in the discussion phase by proposing substantial changes to the wording of their contributions and introduction. However, it is likely that the original claim inflated the scores of other reviewers who noted it as a strength. Broadly, the reviewers appreciated Contribution 2) and the proposed AgentScore method.

Claimed Contribution 3) appears true based on the empirical evaluation, which was mostly appreciated by reviewers. However, for the reader, it would have been interesting to know how stronger unconstrained methods perform on these tasks (e.g., in Table 3), to assess the distance to the performance ceiling. A large part of the paper's argument hinges on the importance of "deployable" models, contrasted with "interpretable", as well as other properties. The authors' definition of "deployable" includes the condition "Evaluation requires no specialized computational infrastructure", which is *highly* debatable in my view. Healthcare regularly adopts some of the most advanced technologies in society, including MRI machines, remote surgery equipment, etc. Surely, these would be considered specialized infrastructures. Moreover, "specialized computational infrastructure" carries almost no meaning. There are huge classes of models that are left out of the empirical comparison that are possible to run on a standard phone, laptop, or desktop computer. Granted, these are not "clinical scoring systems" in the definition of the authors, but this is also possible to challenge.

Overall, the work is solid and immaculately presented, but would benefit from carrying out the improvements suggested by the reviewers.